# Spatial and temporal tracking of cardiac exosomes in mouse using a nano-luciferase-CD63 fusion protein

Weijia Luo[1], Yuan Dai[1], Zhishi Chen[1], Xiaojing Yue[2], Kelsey C. Andrade-Powell[1] & Jiang Chang[1]✉

Exosomes are secreted extracellular vesicles with lipid bilayer membranes. They are emerging as a new category of messengers that facilitate cross-talk between cells, tissues, and organs. Thus, a critical demand arises for the development of a sensitive and non-invasive tracking system for endogenous exosomes. We have generated a genetic mouse model that meets this goal. The *Nano-luciferase* (*NanoLuc*) reporter was fused with the exosome surface marker *CD63* for exosome labeling. The cardiomyocyte-specific *αMHC* promoter followed by the loxP-STOP-loxP cassette was engineered for temporal and spatial labeling of exosomes originated from cardiomyocytes. The transgenic mouse was bred with a tamoxifen-inducible Cre mouse (Rosa26Cre-ERT2) to achieve inducible expression of CD63NanoLuc reporter. The specific labeling and tissue distribution of endogenous exosomes released from cardiomyocytes were demonstrated by luciferase assay and non-invasive bioluminescent live imaging. This endogenous exosome tracking mouse provides a useful tool for a range of research applications.

[1] Center for Genomic and Precision Medicine, Texas A&M University, Institute of Biosciences and Technology, Houston, TX 77030, USA. [2] Department of Obstetrics and Gynecology, Nanfang Hospital, Southern Medical University, 510515 Guangzhou, China. ✉email: jiangchang@tamu.edu

Exosomes are 30–150 nm extracellular vesicles containing DNA, RNA, and proteins. They exist in most organs and body fluids, including blood, tears, urine, amniotic fluid, breast milk, etc[1–3]. The profiles of DNA, RNA, and proteins contained in exosomes are distinct from the exosome-releasing cells[4]. Recent studies revealed that different pathological conditions or stimuli were able to alter exosome sorting preferences, which resulted in the changes of phenotypes or functions in exosome recipient cells or organs[1,5]. Exosomes are emerging as a new category of messengers for intercellular and inter-organ communication and as potential drug-delivery vehicles[6–8]. Therefore, there is a critical need to develop a mouse model for labeling and monitoring of endogenous exosomes. Tracking the biodistribution of endogenous exosomes and kinetics of endogenous exosome release and uptake may have a significant impact on basic and translational studies of exosome-mediated pathways.

Exosomes arise from the endosomal pathway through the invagination of the plasma membrane[1–3,9]. Proteins sorted by the endosomal sorting complexes required for transport (ESCRT) pathway, together with other molecules (such as miRs), are budded into the endosomal membrane to form multivesicular bodies (MVB)[1–3,9]. Exosomes are formed and released to the extracellular matrix upon MVB fusion with the plasma membrane[1–3,9,10]. Therefore, exosomes have lipid bilayer membranes similar to the donor cell plasma membrane carrying major histocompatibility complexes (MHC)[11].

Based on the exosome membrane structure, two major strategies have been applied for exosome labeling: by lipophilic reagent-conjugated reporter that incorporates into the exosome membrane, and by exosome surface marker fusion reporter proteins[12–18]. However, several challenges to monitoring endogenous exosomes currently remain due to the limitation of these exosome-labeling techniques. To generate membrane-conjugated lipophilic reagents, radioisotopes, magnetic nanoparticles and fluorescent dyes are often used as reporters conjugated to a lipophilic reagent[16,17,19–22]. A simple and straightforward labeling procedure links the markers directly to the membrane[16,17,19–22]. However, the mechanism is not specific to exosomes. These methods can only be applied to purified exosomes for in vitro modification, which may not faithfully represent the concentration, location, and dynamic nature of endogenous exosome trafficking. Another approach employs membrane-fused reporter protein labeling[16,17,19–21]. A reporter protein, such as a fluorescent protein or luciferase, is fused with one of the exosome surface markers to label exosomes[16,17,19–22]. Commonly used exosome surface markers are tetraspanins, including CD63, CD9, and CD81[16,17,19–24]. Exosomes released from the donor cells can be tracked by the reporter protein. Current limitations of the fluorescent protein approach include a weak signal and a high-auto-fluorescent background, making non-invasive detection and quantification by fluorescence imaging in vivo difficult[20].

Recently, two studies attempted to track endogenous exosomes in genetic rat and zebrafish models. In the rat model, the fluorescent CD63-GFP fusion was expressed under the control of human Sox2 promoter to label neural stem cell (NSC)-specific exosomes. Unfortunately, this study only demonstrated the exosome uptake in vitro in co-culture conditions[25]. In the zebrafish model, endogenous exosomes were faithfully visualized by the CD63pHluorin fluorescent reporter[19,26]. Thus, a mammalian model for endogenous exosome tracking is currently not available but is critically needed for exosome biology studies.

In this study, we chose NanoLuc as a reporter. NanoLuc is more stable and smaller in molecular size (19 kD) and generates a 150-fold stronger signal compared to traditional Firefly and Renilla luciferases[27]. The half-life of the NanoLuc luminescence signal is >2 h, the longest amongst all known luciferases[27]. Its

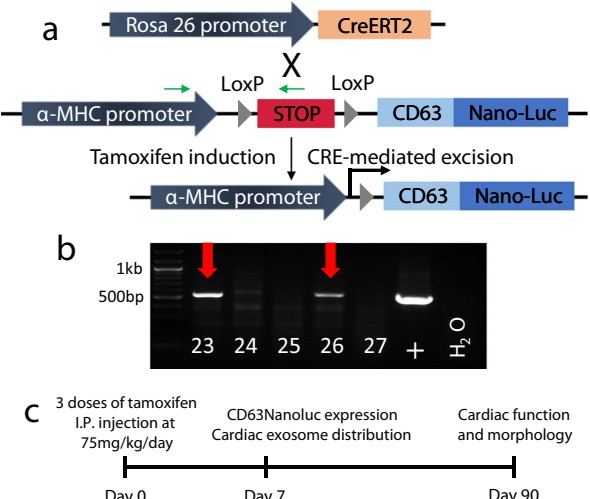

**Fig. 1 Establishment of an inducible cardiac-specific exosome-tracking mouse model. a** Breeding strategy of R26CreERT2; TG-αMHC-STOP-CD63NanoLuc mice. Green arrows indicate genotyping primers. **b** Genotyping PCR of the transgene. #23 and #26 are the two positive founders. **c** Experimental design and timeline.

ultra-stable and highly sensitive signal makes it an ideal reporter for endogenous exosome labeling that can achieve a safe, non-invasive, and quantifiable exosome-tracking objective.

We exploited these NanoLuc properties by fusing it to CD63 and generated a proof-of-concept mouse model to enable tracking of endogenous exosomes. To achieve spatial control of exosome labeling, CD63NanoLuc reporter expression was controlled by the tissue-specific promoter in animal heart. Endogenous exosomes released from the transgenic mouse cardiomyocytes were labeled and tracked. Moreover, the STOP-CD63NanoLuc cassette was introduced to achieve inducible expression of the CD63NanoLuc reporter in vivo upon tamoxifen treatment, allowing temporal control of exosome labeling and tracking, and making such mice more versatile to satisfy diversified research goals.

## Results

**Generation of the inducible cardiac exosome-tracking mice**. To enable non-invasive and quantifiable exosome-labeling, NanoLuc luciferase was fused with the C-terminal of exosome marker CD63 (Fig. 1a). NanoLuc is a small (19 kDa) and the most stable luciferase with a half-life of >2 h. Its luminescent signal intensity is 150-fold stronger than those of Firefly and Renilla luciferases[27]. In order to achieve a controllable exosome tagging, a STOP sequence flanked by two loxP sites was introduced upstream of CD63NanoLuc, and the expression cassette was driven by cardiomyocyte-specific α-MHC promoter (Fig. 1a). Linearized DNA containing αMHC-STOP-CD63NanoLuc cassette was used for pronuclear microinjection. DNA extracted from mouse tails was used for genotyping purposes. The TG-αMHC-STOP-CD63NanoLuc mouse lines were successfully established and confirmed by PCR genotyping (Fig. 1b).

**Expression of CD63NanoLuc reporter was tightly controlled**. To validate the spatial and temporal induction of exosome reporter CD63NanoLuc expression, TG-αMHC-STOP-CD63NanoLuc transgenic mice were bred with the tamoxifen-inducible Cre mice, Rosa26Cre-ERT2 (R26CreERT2). This resulted in inducible cardiomyocyte-specific R26CreERT2;TG-αMHC-STOP-CD63NanoLuc transgenic mice. The STOP cassette was deleted by Cre-ERT2 recombinase upon tamoxifen induction (Fig. 1a).

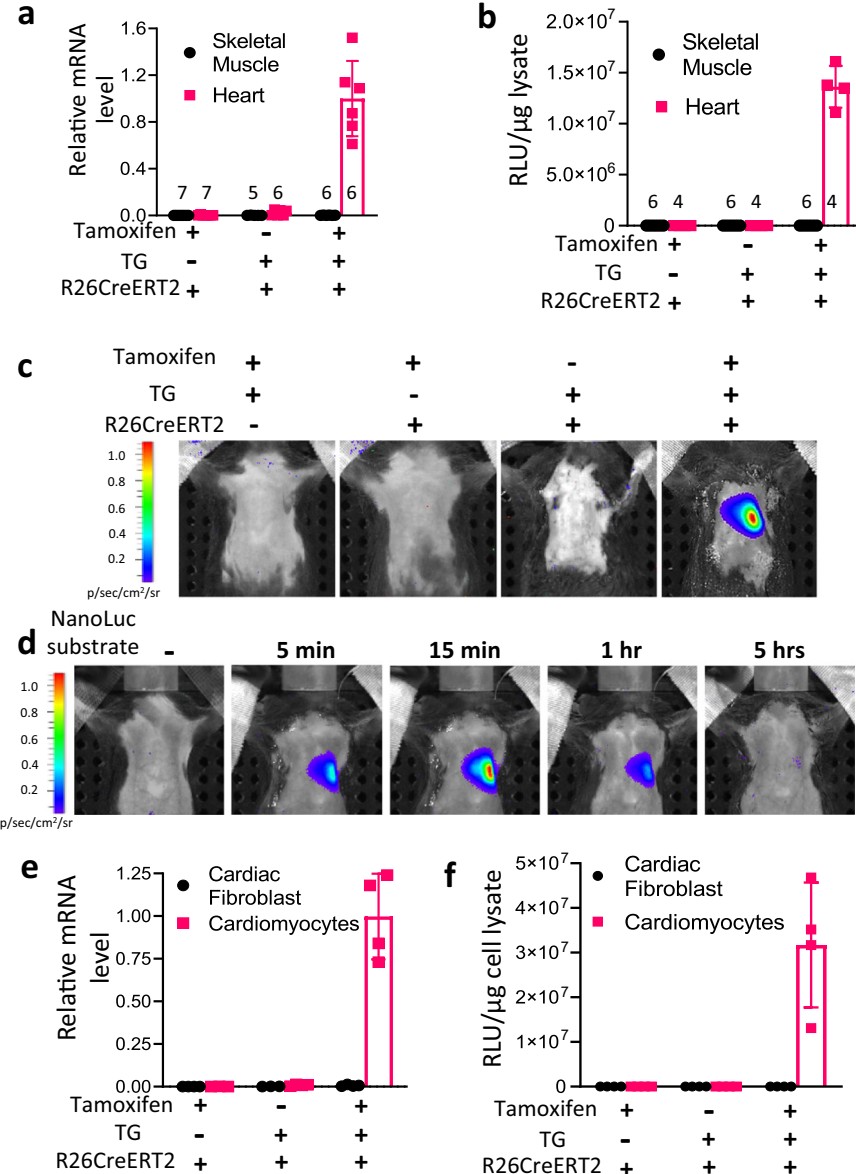

**Fig. 2 CD63NanoLuc expression is precisely controlled. a, b** CD63NanoLuc transcript and luciferase activity were detected only in R26CreERT2; TG-αMHC-STOP-CD63NanoLuc mouse heart upon tamoxifen induction. Sample sizes are indicated on each column. $p < 0.0001$. **c** Bioluminescence imaging was shown within the heart region in tamoxifen-induced R26CreERT2; TG-αMHC-STOP-CD63NanoLuc mice. **d** Time course of the bioluminescence signal in R26CreERT2; TG-αMHC-STOP-CD63NanoLuc mouse. **e, f** CD63NanoLuc transcript and luciferase activity were detected only in cardiomyocytes isolated from tamoxifen-treated R26CreERT2; TG-αMHC-STOP-CD63NanoLuc mice. $p < 0.0001$. TG TG-αMHC-STOP-CD63NanoLuc. $n = 4$ for **e–f**. All bar graph expressed as mean ± SD.

CD63NanoLuc expression was induced by intraperitoneal (I.P.) injection of tamoxifen at $75\,mg\,kg^{-1}$ body weight/day for 3 days. CD63NanoLuc expression and cardiac exosome labeling were assessed 1 week after tamoxifen induction. To determine the effects of long-term overexpression of CD63NanoLuc on the heart, cardiac function and morphology were analyzed 3 months after tamoxifen induction (Fig. 1c). To validate the spatially and temporally controlled exosome marker expression, we assessed the mRNA and protein levels of the transgene in the tissue lysate by quantitative PCR and luciferase assays, respectively. As expected, upon tamoxifen induction, a robust CD63NanoLuc transcript was detected only in the heart (Fig. 2a and Supplementary Fig. 1a). Consistent with mRNA expression, a strong luciferase activity was detected post-induction exclusively in the transgenic heart (Fig. 2b and Supplementary Fig. 1b).

Next, we assessed CD63NanoLuc expression in live animals by IVIS SpectrumCT, a non-invasive bioluminescent imaging system. To induce the bioluminescence signal, 100 µl NanoLuc substrate was delivered to the animal via intraperitoneal (I.P.) injection. Sterile I.P. injection of PBS was served as a control. The luminescence signal was detected by live imaging 5 min after substrate injection. Upon tamoxifen induction, a strong bioluminescence signal was exhibited only in the heart region of transgenic animal (Fig. 2c). The bioluminescence signal was detected as early as 5 min and peaked at 15 min after the substrate injection. The signal was continuously detected and diminished 5 h after the substrate injection (Fig. 2d). These results suggest that the NanoLuc bioluminescence is highly sensitive, stable and detectable non-invasively in live mice.

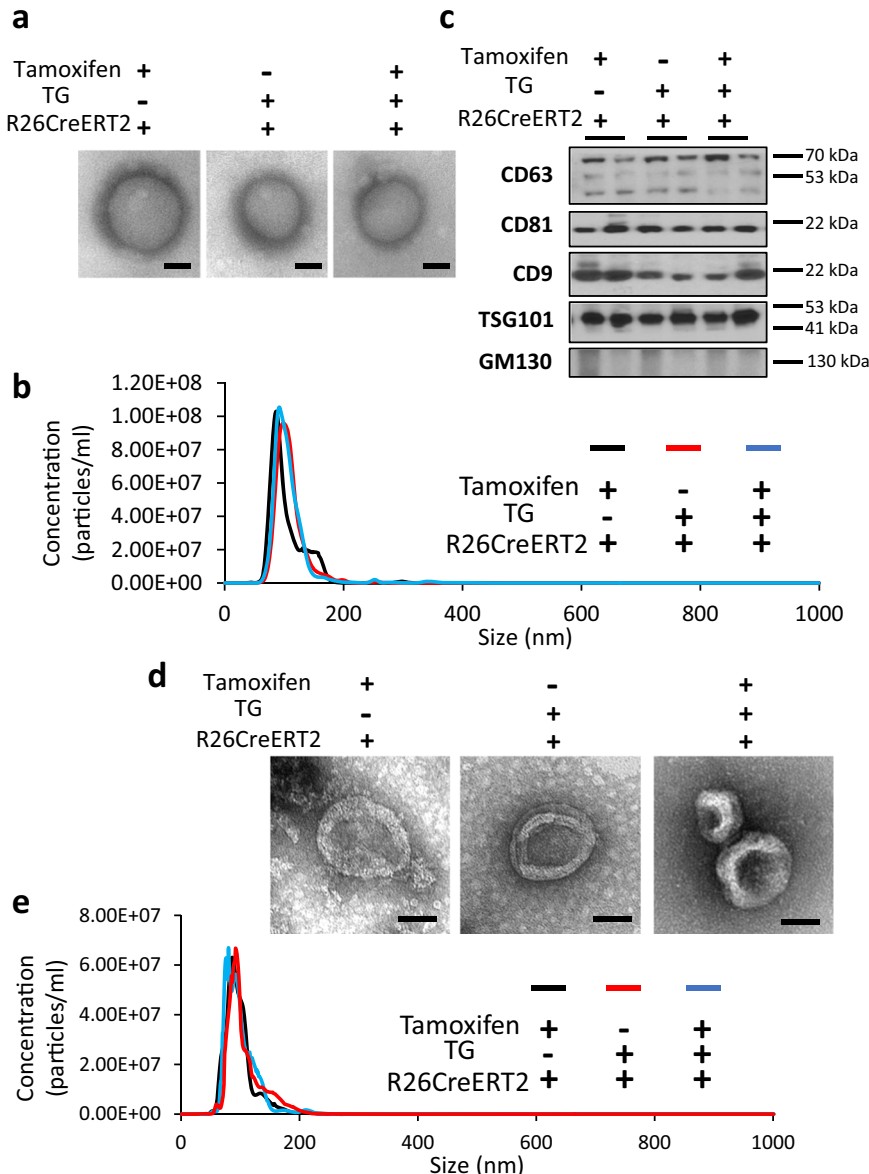

**Fig. 3 Evaluation of the quality of exosomes. a** Morphological appearances of plasma exosomes shown by transmission electron microscopy. **b** Size distribution of plasma exosomes assessed by nanoparticle tracking analysis (NTA). **c** Western blot of plasma exosome markers. **d** Morphological appearances of cardiomyocyte-derived exosomes shown by transmission electron microscopy. **e** Size distribution of cardiomyocytes releasing exosomes assessed by NTA. TG TG-αMHC-STOP-CD63NanoLuc.

We then evaluated the cell specificity and leakage of the inducible system. The expression levels of CD63NanoLuc mRNA and protein were assessed in the primary cardiomyocytes and fibroblasts, the two major cell types in the heart. We found that tamoxifen treatment induced CD63NanoLuc mRNA and protein only in cardiomyocytes but not in cardiac fibroblasts (Fig. 2e, f), suggesting that this exosome-tracking system was highly cell-specific and tightly controlled.

**Validation of exosome isolation and purification.** To validate exosome labeling and detection by bioluminescence, the trans-genic and control mice were given tamoxifen by I.P. injection, with vehicle (peanut oil) injections as controls. One week after induction, exosomes were isolated from mouse plasma and primary cell culture supernatant by Exoquick[TM] precipitation and ultracentrifugation methods, respectively. We evaluated the

quality of exosomes by morphology, size and molecular markers. The transmission electron microscopy displayed exosomes as vesicles with membrane structures and sizes in the 50–150 nm range (Fig. 3a, d). Overexpression of CD63NanoLuc did not affect exosome morphology and size (Fig. 3a, d, right panel). The overall size distribution of exosomes was assessed by nanoparticle tracking analysis (NTA), showing particle size fall in the 30–150 nm range (Fig. 3b, e). No significant differences of exo-some concentration and size distribution were detected when comparing the NanoLuc-labeled exosomes to the controls (Fig. 3b, e). The expression levels of four exosome surface mar-kers, CD63, CD81, CD9, and TSG101, were assessed by western blot[3,23]. GM130, a marker of membrane-bound vesicles from Golgi apparatus, was used in the immunoblot assessment as a non-exosome negative control[23] (Fig. 3c). All exosome surface markers but not GM130 were detected in the NanoLuc-labeled exosomes and the control groups. All assessments confirmed the

quality of exosomes, and that CD63NanoLuc overexpression had no deleterious effects on exosome synthesis, morphology, and release.

**Tracking and quantifying of cardiac exosomes among organs.** To demonstrate that cardiomyocyte-derived exosomes were labeled with NanoLuc, exosomes isolated from primary cardiomyocytes and cardiac fibroblasts were analyzed by luciferase assay. A strong bioluminescence signal was exhibited in the exosomes released from the tamoxifen-induced transgenic cardiomyocytes. No luciferase activity was detected in the transgenic cardiomyocytes without tamoxifen induction or cardiac fibroblasts with or without tamoxifen induction (Fig. 4a), indicating that cell-specifically released endogenous exosomes can be labeled with a bioluminescent reporter. The induction of this cardiomyocyte-specific exosome-tracking system was highly efficient and tightly controlled.

Next, we measured the luciferase activity in the exosomes isolated from animal plasma. Strong bioluminescence was detected in transgenic mice with tamoxifen induction but not in plasma from transgenic mice without tamoxifen induction or in R26CreERT2 mice (Fig. 4b), indicating that exosomes derived from cardiomyocytes were released into blood. This result provided clear evidence supporting inter-organ communication via blood circulation of exosomes.

To reveal the biological distribution profile of endogenous exosomes specifically released from cardiomyocytes, we collected thirteen mouse organs and compared luciferase activity among them. We found that cardiac exosomes were mainly uptaken by thymus, testis, lung and kidney, as shown by strong bioluminescent signals. Low-luciferase activity was detected in spleen and bone marrow cells, and minimum luciferase activity could be detected in skeletal muscle, thyroid, white adipose tissue, intestine, liver, pancreas and brain (Fig. 4c). Without tamoxifen induction, luciferase activity was not detectable in these remote organs and tissues (Supplementary Fig. 1b). These results unveiled a unique tissue preference of the exosomes produced from cardiomyocytes.

To further demonstrate that the NanoLuc tissue distribution shown in Fig. 4c represents the endogenous cardiac exosome behavior, we analyzed and compared free NanoLuc tissue distribution. Free NanoLuc with $1 \times 10^7$ Relative Luciferase unit (RLU), which is comparable to the total plasma RLU in the CD63NanoLuc-overexpressing mice, was introduced into wild-type animals via tail vein injection. The luciferase activity in mouse plasma was followed 3, 6, 12, and 24 h after injection. Reduction to 2.4% and 4‰ of the starting RLU was detected 3 and 6 h after injection, respectively, indicating rapid degradation of free NanoLuc in circulation. Twenty-four hours after injection, the plasma luciferase activity returned to the basal level (Supplementary Fig. 2a). Next, we assessed the 12-h NanoLuc tissue distribution by luciferase assay. Bioluminescence signals were mainly detected in liver but not in testis and kidney (Supplementary Fig. 2c), which is significantly different from the cardiac exosome-associated NanoLuc distribution. Furthermore, we used a 100-kDa protein concentrator to separate the exosome-conjugated CD63NanoLuc from exosome-free NanoLuc, which include free NanoLuc (19 kDa) and free CD63NanoLuc (45 kDa), in the mouse plasma. We found that 93% of luciferase activity in plasma was due to exosome-associated CD63NanoLuc and 7% was due to exosome-free NanoLuc (Supplementary Fig. 2c). Taken together, these results clearly indicate that the reporter mouse faithfully tracks cardiac exosomes in vivo.

**Assessment of exosome-mediated intercellular communications.** To demonstrate intercellular communications via exosomes,

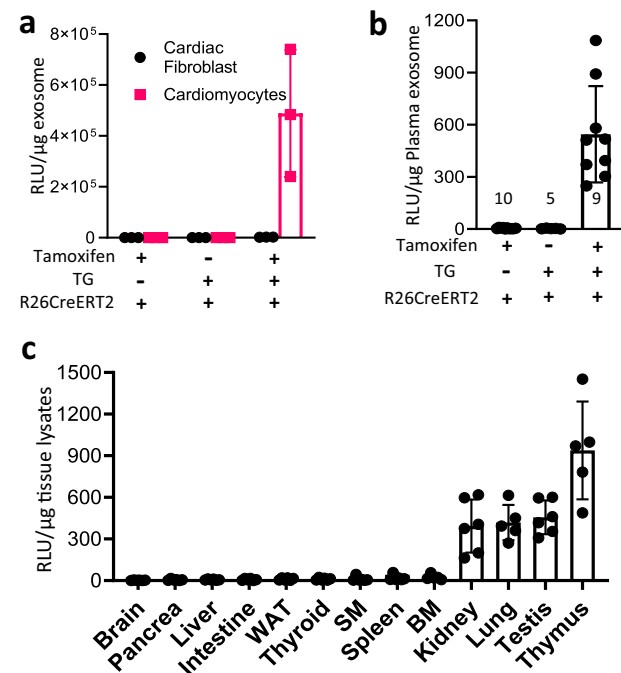

**Fig. 4 Assessment of cardiomyocyte-derived exosome labeling and tissue distribution. a** A strong luciferase activity was detected only in exosomes released from cardiomyocytes isolated from tamoxifen-treated R26CreERT2; TG-αMHC-STOP-CD63NanoLuc mice. $p = 0.0045$, $n = 3$. **b** An obvious luciferase activity was shown in plasma exosomes from the same mouse cohort. Sample sizes are indicated on each column. $p < 0.0001$. **c** Tissue uptake of cardiomyocyte-derived exosomes were detected by luciferase activity in 13 organs. WAT: White adipose tissue. SM Skeletal muscle. BM Bone marrow. $n = 6$. TG TG-αMHC-STOP-CD63NanoLuc. $p < 0.0001$. All bar graph expressed as mean ± SD.

isolated wild-type cardiac fibroblasts were treated for 24 h with CD63NanoLuc-labeled exosomes isolated from the supernatant of CD63NanoLuc transgenic cardiomyocyte culture. A profound increase in luciferase activity was observed in the cardiac fibroblasts (Fig. 5a), suggesting the uptake of tagged exosomes by cardiac fibroblasts. Next, we evaluated exosome-mediated cardiomyocyte and cardiac fibroblast communication in vivo. One week after tamoxifen induction in CD63NanoLuc transgenic mice, cardiac fibroblasts were isolated from the heart followed by luciferase activity assessment in the cells. A strong luciferase activity was detected in cardiac fibroblasts, showing the occurrence of paracrine interaction of cardiomyocytes and cardiac fibroblasts via exosomes (Fig. 5b).

**Expression of CD63NanoLuc had no effect on the heart.** To investigate if the long-term expression of CD63NanoLuc reporter protein had any deleterious effect on the mouse heart, we evaluated animal hearts by morphology and cardiac function 3 months after tamoxifen induction. The results were compared among the following six groups of mice: wild-type, TG-αMHC-STOP-CD63NanoLuc, together with R26CreERT2 and R26CreERT2;TG-αMHC-STOP-CD63NanoLuc with or without tamoxifen induction. The ratios of heart weight over body weight and heart weight over tibia length were measured (Supplementary Fig. 3a, b). Cardiac functions were evaluated by echocardiographic analysis (Supplementary Fig. 3c). The heart size and morphology were analyzed by whole mount images and H&E staining (Supplementary Fig. 3d, e). All assessments showed no significant differences between the control groups

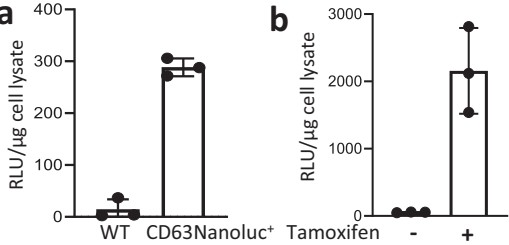

**Fig. 5 Demonstration of paracrine interaction of exosomes from cardiomyocytes with cardiac fibroblasts in vitro and in vivo. a** Luciferase activity was evaluated in the wild-type cardiac fibroblasts treated with unlabeled (WT) and CD63NanoLuc-labeled exosomes, respectively. The CD63NanoLuc-tagged exosomes were collected from the culture supernatant of primary cardiomyocytes isolated from tamoxifen-treated R26CreERT2; TG-αMHC-STOP-CD63NanoLuc mice. $p < 0.0001$. $n = 3$. **b** A substantial luciferase activity was detected in the cardiac fibroblasts isolated from tamoxifen-treated R26CreERT2; TG-αMHC-STOP-CD63NanoLuc mice. $p = 0.0047$. $n = 3$. All bar graph expressed as mean ± SD.

and the induced group (Supplementary Fig. 3). Altogether, these results suggest that long-term expression of CD63NanoLuc has no influence on CD63 biological functions in the heart.

## Discussion

There is a technological challenge and demand for a sensitive and non-invasive tracking system for endogenous exosomes. Such a system may provide a powerful tool for the exploration of the biological functions, mechanisms, and clinical applications of exosomes in a broad spectrum of research. In this study, we have met such a demand by generating a proof-of-concept genetic mouse that enables tracking of endogenous exosomes specifically released from cardiomyocytes. This mouse model is superior to current ex vivo and in vivo labeling approaches and overcomes current barriers to exosome monitoring. This model provides a mouse tool for the exploration of biological functions and clinical applications of exosomes in cardiovascular diseases. Moreover, with this proof-of-concept study, the STOP-CD63NanoLuc can be used in an even broader spectrum of applications in basic and translational life sciences and biomedical studies.

Three categories of reporters have been selected to visualize and track exosome. The first reporter category includes radioisotopes, magnetic nanoparticles or fluorescent dyes. To label exosomes, reporters are directly linked to the membrane by conjugation to lipophilic reagents[17–20]. The procedures are relatively easy and straightforward. Radioisotope and magnetic resonance monitoring involves strong signals that can be detected deep in live animals[15,28]. However, almost all of them rely on the administration of exogenously labeled exosomes at an extremely high concentration, which can be two-fold higher than endogenous serum exosome concentration in order to be detected[21]. This disproportionate exosome loading may trigger deleterious effects in vivo. Carrying the donor MHC, exogenous exosomes can be targeted and cleared by the recipient animal's immune system. The exosome behavior or biological function in vivo might be jeopardized or altered due to contamination, low recovery, or aggregation of the exosomes introduced during exosome isolation or preparation. Important paracrine and autocrine effects exhibited by exosomes could be lost as a result. Therefore, it is not surprising to see that most studies using this strategy have resulted in a similar tissue-distribution pattern of exosomes. The injected exosomes are heavily accumulated in the liver, spleen and lungs regardless of the exosome origin[16,20], which does not faithfully represent tissue deposition of endogenous exosomes. Our mouse model shows no significant amount of cardiac exosomes in the liver and spleen under physiological conditions. An artificial effect of exogenous exosome delivery was also implied by a study, which reported that the same exosomes can exhibit different trafficking patterns when administered orally or by intravenous injection[29], indicating that a single delivery method of exogenous exosomes may lead to a misleading exosome trafficking pattern.

Fluorescent protein labeling is accomplished by fluorescent protein fused to either palmitoylation signals or exosome surface markers, such as CD63[14,24,30]. A recent elegant study demonstrated a successful application of this strategy to labeling and tracking of endogenous exosomes in zebrafish. The unique transparent body makes zebrafish an ideal model for the visualization of a fluorescent signal[26]. However, the low signal level and high-auto-fluorescent background make the luminescence signal very difficult to be reliably detected and quantified non-invasively in mice[20]. In another recent study, a genetic rat model of Sox2 promoter-driven CD63-GFP has been generated for exosome tracking. However, the exosome labeling and transfer was assessed only in primary cells in vitro[25], probably due to the too low signal level for in vivo detection.

Luciferase labeling is a promising technology for tracking endogenous exosomes. Gaussia luciferase (gLuc) fused to lactadherin or PDGFR has been reported to label exosomes[31–33]. It overcomes the auto-fluorescent background of fluorescent proteins and could be potentially detected in live animals by a non-invasive bioluminescent imaging system, allowing signal quantification and endogenous exosomes tracking[34]. However, the gLuc luminescence has an extremely short half-life of 2 min[35], which limits its in vivo application in mice.

The unique chemical characteristics of NanoLuc bioluminescence make it the top reporter selection for in vivo tracking of endogenous exosomes in mice. The half-life of the NanoLuc luminescence signal is >2 h, the longest amongst all known luciferases[27]. In our transgenic mice, the bioluminescence signal remains detectable in vivo by IVIS SpectrumCT In Vivo Imaging System even 1 h after the substrate injection, demonstrating a reliable and ultra-sensitive mouse model for detecting endogenous exosomes. The CD63NanoLuc-labeled cardiac exosomes are functional and could be uptaken by recipient cells in vivo and in vitro. Transmission electron microscopy and NTA analyses have also demonstrated that NanoLuc fusion to CD63 has no deleterious effects on cardiac exosome biogenesis. CD63NanoLuc has been recently used in human cancer cell lines for exosome tracking[10]. CD63NanoLuc luciferase activity was well correlated to NTA data under various conditions. These CD63NanoLuc-labeled exosomes were functional and could be uptaken in vitro, and in vivo in the xenograft tumor model, indicating that CD63NanoLuc fusion protein did not affect exosome biogenesis and functions[10], consistent with our results.

Another excellent recent study used Cre recombinase as a sensor to show extracellular vesicle (EV) trafficking between implanted exogenous donor cells (expressing Cre) and recipient cells (expressing fluorescent protein) in mice. Protein fluorescence in the recipient cells was switched from red to green upon Cre delivery by EVs[36,37]. While the study provided the proof-of-concept evidence of functional EV transport in vivo, it required transplantation of exogenous cells (exosome donor cells) and did not represent a true endogenous EV tracking system. With a green fluorescence protein providing the readout, this model carries the drawbacks of fluorescent protein labeling previously discussed and is not quantitative due to the permanent gene switch. Moreover, the Cre mRNA was not specifically loaded into exosomes. Other vesicles with lipid structure, such as apoptotic bodies and microvesicles, may not be distinguishable.

Since we chose the animal heart to demonstrate the spatial tracking of the exosomes released from cardiomyocytes, the cardiac exosome studies would greatly benefit from this animal model. A growing body of evidence suggests that exosomes are involved in the regulation of cardiomyocyte survival, injury healing, fibrosis, angiogenesis and inflammation[38–45]. Cardioprotective exosome delivery has been considered as a new treatment strategy due to their modulation capability, delivery specificity and uptake efficiency[46–50]. As a new category of messengers, exosomes have been found to facilitate cross-talk between cardiomyocytes and non-cardiomyocytes. However, this exosome-mediated cellular communication was only demonstrated in vitro due to the lack of a sensitive and faithful mouse model[38]. Currently, the identification of exosome donor cells is also challenging and difficult. One recent study showed that the exosomes released from an infarcted heart promoted injury of the transplanted bone marrow mesenchymal stem cells[51]. However, it was impossible to know what types of exosome donor cells were responsible for this deleterious consequence. Our mouse model provides a useful tool to achieve these goals and allows us to track the dynamics of the exosomes derived from cardiomyocytes in vivo.

This mouse model also provides quantification of the endogenous exosome uptake via both paracrine and endocrine mechanisms. We found that cardiomyocyte-released exosomes were delivered to cardiac fibroblasts ($2.2 \times 10^3$ RLU $\mu g^{-1}$, Fig. 5b), as well as to remote organs ($3-9 \times 10^2$ RLU $\mu g^{-1}$, Fig. 4c), suggesting that exosome trafficking is more substantial in a paracrine manner than in an endocrine manner. Consistently, cardiomyocyte-released exosomes only comprise 1.1‰ of the plasma exosomes, calculated by the RLU in plasma exosomes (Fig. 4b) over the RLU in cardiomyocytes-released exosomes (Fig. 4a). With the low level of cardiac exosomes in circulation, it was not surprising that the weak signal in remote organs was beyond the limit of imaging detection. Even with this limitation, this model still enables us to measure a precise amount of cardiac exosomes uptaken by the animal organs and even cells through the assessment of their luciferase activities, providing the valuable tissue-distribution profile for endogenous exosomes.

The TG-αMHC-STOP-CD63NanoLuc mouse is the first genetic mouse model capable of tracking endogenous exosomes. The LoxP-flanked STOP sequence enables the inducible expression of the CD63NanoLuc reporter in vivo, allowing temporal control of exosome labeling and tracking. Therefore, it provides a valuable tool for the studies of exosome communication under different physiological and pathological conditions. Moreover, CD63NanoLuc-tagged exosomes are more specific, easier to detect and quantify compared to fluorescently-labeled exosomes. Finally, when bred with gene-specific floxed mice, this model can be used to study gene-specific exosome functions.

It should be noted that while CD63 is a well-recognized exosome marker, CD63 may not label exosomes under certain extreme pathological conditions, such as necrosis. It was also reported that exosomes secreted from certain types of cancer cells do not express common markers like CD63[52–54]. Even so, CD63 remains one of the most widely used exosome markers in both physiological and pathological conditions. The current mouse model should cover most of research tasks in the exosome research field.

## Methods

**Animals**. Generation of the TG-αMHC-STOP-CD63NanoLuc mice was described in results. The BamHI-PacI DNA fragment of αMHC-STOP-CD63NanoLuc was used for pronuclear microinjection in a C57/B6 background. Genomic DNA was isolated from mouse tails for genotyping. PCR primers for genotyping were: Forward: GAAGTGGTGGTGTAGGAAAG; Reverse: GTCACACCACAGAAGTA AGG. To induce CD63NanoLuc expression, three dosages of tamoxifen or peanut oil (as a vehicle control) were injected intraperitoneally to 8–10-week-old adult

mice at 75 mg kg$^{-1}$ body weight. Cardiac function, exosome labeling, and distribution were assessed 1 week or 3 months after induction. B6.129-Gt(ROSA) 26Sortm1(cre/ERT2)Tyj/J (R26CreERT2) mouse line was purchased from The Jackson Laboratory (Bar Harbor, ME, USA). All animal experiments were conducted in accordance with the protocol approved by the Institutional Animal Care and Use Committee of the Texas A&M University Health Science Center-Houston.

**Quantitative PCR analysis**. mRNA expression levels were determined by quantitative PCR (qPCR) (StepOnePlus, ThermoFisher Scientific, MA, USA) as described previously[55]. Total RNA was extracted using Direct-zol RNA kits (R2053, Zymo Research, CA, USA). cDNA was synthesized by qScript Flex cDNA kit (95047, Quantabio, MA, USA). PCR primers were as follows: mGAPDH Forward: CATGGCCTTCCGTGTTCCTA; Reverse: CCTGCTTCACCACCTTCTTGAT. NanoLuc Forward: GTCCGTAACTCCGATCCAAAG; Reverse: GTCACTCCG TTGATGGTTACTC.

**Luciferase assay**. For cardiomyocyte and cardiac fibroblast samples, cells in 12-well plates were lysed by 100 μl Passive Lysis Buffer (E1941, Promega, CA, USA)/well. Forty microliters of cell lysate was reacted with 20 μl NanoLuc substrate (N1110, Promega, CA, USA) for bioluminescence reading (Lumat3, LB9508, Berthold, Germany). For tissue samples, 10 mg tissue sample was lysed by 200 μl Passive Lysis Buffer. Tissue lysate (100 μl), except for heart, was reacted with 20 μl NanoLuc substrate for bioluminescence reading. For luciferase assay, 100 μl of 1:1,000 diluted heart tissue lysates were used. For exosome sample, 16 μl exosome suspension was lysed by 4 μl 5x Passive Lysis Buffer and reacted with 20 μl NanoLuc substrate. Luciferase activity of each sample was measured three times. All results were normalized to cell/tissue/exosome lysate protein concentration.

**Non-invasive in vivo bioluminescence imaging**. In vivo bioluminescence imaging of mice was done with an IVIS SpectrumCT In Vivo Imaging System (128201, PerkinElmer, CT, USA). For this, 100 μl of 1:10 diluted NanoLuc substrate (N1110, Promega, CA, USA) was administrated into mice via intraperitoneal injection to produce bioluminescence in vivo. Bioluminescence images were taken 5 min after substrate injection with 5 min of exposure time.

**Isolation of primary adult cardiomyocytes and fibroblasts**. Mouse cardiomyocytes and cardiac fibroblasts were isolated from 12–20-week-old adult male mice. Cardiomyocytes were digested from the heart by adult rat/mouse cardiomyocyte isolation kit (ac-7031, Cellutron, MD, USA) with Langendorff perfusion system (120108, Radnoti, CA, USA). Cells were cultured for 72 h in the laminin-coated dishes with AW cardiomyocyte culture medium without serum (m-8034, Cellutron, MD, USA). RNA and total protein were then collected for NanoLuc expression analysis. The cell culture supernatants were collected for exosome isolation.

Cardiac fibroblasts were isolated by enzymatic digestion with Langendorff perfusion system. The enzyme cocktail digestion perfusion buffer contained 100 U ml$^{-1}$ Collagenase II (17101015, ThermoFisher Scientific, MA, USA) and 0.1% Trypsin (20233, ThermoFisher Scientific, MA, USA). Cells were seeded with DMEM medium (CM002, GenDEPOT, TX, USA) containing 10% exosome-depleted FBS (A2720801, ThermoFisher Scientific, MA, USA) and 100 U ml$^{-1}$ PenStrep. Sixteen hours later, cardiac fibroblasts were collected for RNA and protein analysis. To assess the primary cardiac fibroblast exosome uptake, cells were treated with NanoLuc-labeled exosomes derived from cardiomyocytes for 24 h followed by luciferase assay in the treated cardiac fibroblasts.

**Exosome isolation**. Plasma exosomes were isolated from EDTA-treated mouse plasma. The plasma samples were treated with 5 U ml$^{-1}$ thrombin (605157 Millipore Sigma, MA, USA) and filtered through 0.22 μm polyvinylidene fluoride (PVDF) membrane. Then, 50.4 μl Exoquick$^{TM}$ (EXOQ5A-1, System Biosciences, CA, USA) was added to 200 μl plasma sample to precipitate exosomes as described in the manufacturer's protocol. Excess Exoquick$^{TM}$ was removed by passing samples through PD SpinTrap$^{TM}$ G-25 column (28-9180-04, GE Healthcare, Buckinghamshire, UK).

Cardiomyocyte- and cardiac fibroblast-derived exosomes were isolated from primary cardiomyocyte and cardiac fibroblast culture media, respectively. Briefly, primary cell culture media was filtered through 0.22 μm PVDF membrane. Then, 200 μl Exoquick-TC$^{TM}$ (EXOTC10A-1, System Biosciences, CA, USA) was added to every 1 ml culture medium to precipitate exosomes. The exosome pellet was re-suspended with 0.1 μm polyethersulfone (PES) membrane-filtered PBS. Excess Exoquick$^{TM}$ was removed by passing samples through PD SpinTrap$^{TM}$ G-25 column (28-9180-04, GE Healthcare, Buckinghamshire, UK). Alternatively, 0.22 μm PVDF membrane-filtered primary cell culture media was further concentrated by 100-kDa protein concentrator (88533, ThermoFisher Scientific, MA, USA). Concentrated culture media was then centrifuged at 100,000 × g for 15 min using Airfuge (347854, Beckman Coulter, IN, USA). Exosome pellet was washed with 0.1 μm PES membrane-filtered PBS and ultracentrifuged again at 100,000 × g for 15 min using Airfuge. Exosome pellets was re-suspended with 0.1 μm PES membrane-filtered PBS for NTA and morphology analyses.

**Transmission electron microscopy.** Freshly prepared plasma exosomes were fixed in 2% paraformaldehyde at 4 °C overnight. Samples were placed on 100 mesh carbon coated, formvar-coated copper grids treated with poly-L-lysine for ~1 h. Samples were then negatively stained with Millipore-filtered aqueous 1% uranyl acetate for 1 min. Stain was blotted dry from the grids with filter paper and samples were allowed to dry[56]. Pictures were taken by JOEL JEM-1010 Transmission electron microscope (JOEL, MA, USA) at a voltage of 80 kV.

**Nanoparticle tracking analysis.** Exosome size distributions were determined by NanoSight (LM10, Malvern Panalytical, MA, USA) with 488-nm laser and NTA3.1 software, according to the manufacturer's protocol. Three 30 s measurements were done for each sample. Plasma exosome samples were diluted 1000-fold in PBS for NTA analysis.

**Western blot.** Exosome protein samples for western blot were extracted by 10xRIPA buffer with protease inhibitors. Antibodies were purchased from the following sources: rabbit anti-CD63 (sc-15363, Santa Cruz, TX, USA); mouse anti-TSG101 (sc-7964, Santa Cruz, TX, USA); mouse anti-CD81 (sc-166029, Santa Cruz, TX, USA); mouse anti-CD9 (sc-13118, Santa Cruz, TX, USA); mouse anti-GM130 (sc-55590, Santa Cruz, TX, USA); anti-rabbit IgG HRP-linked (7074S, Cell Signaling, MA, USA); anti-mouse IgG HRP-linked (7076 S, Cell Signaling, MA, USA). 1 µg ml$^{-1}$ primary antibodies, and 50 ng ml$^{-1}$ secondary antibodies were used for blotting.

**Statistics and reproducibility.** All experiments were independently repeated at least three times. Numbers of biological repeats are indicated in figures. Data are expressed as mean ± SD. For multiple group comparisons, one-way ANOVA or two-way ANOVA followed by Brown–Forsythe and Bartlett's test was performed. For two group comparisons, unpaired, two-tailed Student's $t$-test was used. A $p$-value $< 0.05$ was considered statistically significant. GraphPad Prism software 8.3 were used for statistical analysis. Experiments and data analysis were conducted double blinded by investigators. Experimental animals were age-matched littermates.

**Reporting summary.** Further information on research design is available in the Nature Research Reporting Summary linked to this article.

## Data availability
The data supporting the findings of this study are available within the paper and its' Supplementary Information. Each data point corresponding to figures that describe the results from in vivo or in vitro model studies are provided as separate Source Data for Fig. 2a, b, e, f, 4a–c, 5a, b and Supplementary Data Figs. 1a, b, 2a–c, 3a–c. Other source data related to the study are available from the corresponding author upon reasonable request.

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

## Acknowledgements

We would like to thank Dr. Vladimir N. Potaman for editorial assistance. This study was supported by American Heart Association Innovative Project Award 18IPA34180012 (J.C.). Transmission electron microscopy was conducted at the M.D. Anderson High-Resolution Electron Microscopy Facility Core that was supported by CCSG grant NIH P30CA016672.

## Author contributions

W.L. and J.C. designed the experiments; W.L. conducted most experiments; Z.C. collected data in Fig. 4 & Supplementary Fig. 2. W.L., Y.D., X.Y., K.C.A., and J.C. analyzed the data; W.L. and J.C. wrote the manuscript.

## Competing interests

The authors declare no competing interests.
