## [Peer Review File · Communications Biology]

Reviewers' comments:

Reviewer #1 (Remarks to the Author):

The manuscript "Spatial and temporal control of endogenous exosome tracking in mouse" by Luo et al. aims to provide a new mouse model for tracking endogenous cardiac exosomes. It builds on the use of NanoLuc that reaches emission 150-fold stronger than the signal emitted by Firefly and Renilla luciferase. Such tool should allow tracking of EVs at the whole-animal scale. Although the aim of this manuscript and research is of great interest to the field, the authors unfortunately fall short in demonstrating that they actually track exosomes.

Main concerns :

- Specificity of CD63-NanoLuc for the tracking of exosomes: In order to fully prove that organ inter-communication is mediated by exosome-mediated CD63-NanoLuc, the authors could compare the behavior of a similar mouse tagged with NanoLuc only (not tagged to CD63). Besides, CD63 has been fused to its Cterminal region known to contain regulatory domains of CD63 : have the authors assessed whether this impact number and morphology of EVs, both in culture and in mouse blood circulation ?
- Along this line, purification of EVs is performed by ExoQuick, which is known to be relatively prone to artefacts. This referee would recommend performing classical ultracentrifugation (and gradients) for ensuring quality of the purified and to avoid the potential contamination of non-EVs CD63-LUC protein or mRNA.
- The manuscript frustratingly lacks a clear demonstration that the tool could be used to track shedding and fate of cardiac exosomes in vivo. In order to validate this approach, one should demonstrate that luciferase activity could be use in whole-animal to track the fate of exosomes shed from the cardiomyocytes. Such information is partly contained in Fig.3c but the benefit of using a brighter form of luciferase to perform imaging in a living animal is not demonstrated.
- The aim of developing a cardiomyocyte-specific transgene isn't clear and not demonstrated in this manuscript – one should demonstrate that exosome released by cardiomyocytes are functional and that such function can be tracked using this new imaging tool.

Minor comments:

- to further ensure specificity and exclude promoter leakiness, overall luminescence signal (photon flux) should be quantified in fig.2c to ensure absence of non-specific expression.
- a similar quantification for the signal obtained in the heart, and the entire animal, after substrate injection should be performed to provide a clear description of the transgene biodistribution and lifetime/half-life
- additional controls and detail of exosome purity and presence should be provided for figure 3a and 3b (see major comments)
- Figure 3c requires additional proof of the specificity of the signal for exosome CD63-NanoLuc : could the authors provide IVIS images or additional proof that exosomes can be found in these organs ? Maybe, providing control images of similar animal/organs in absence of tamoxifen would support the claims. In addition, when the tamoxifen treatment is stopped, how long can the luciferase signal be detected ? Such comparison would allow to assess EVs half-life.
- it is not clear, in Fig.5a, where and how were the exosomes purified. The authors should be more clear, along the manuscript, and carefully detail what has been done.
- data provided in figure 6 are useful but are only control experiments demonstrating that overexpression of this construct does not affect cardiac development. As such, this should be a supplemental information.
- the manuscript would benefit from english proofreading
- the introduction is rather superficial considering the amount of approaches that have been already

published (reviewed in Hyenne et al. CAM 16) and lacks referencing to two recent methods that have been published, in particular using the zebrafish embryo (although one of them is mentioned superficially in the discussion).

- in the discussion, when the authors compare injection of exogenous EVs to endogenous, they should discuss the entry/injection routes.

Reviewer #2 (Remarks to the Author):

Manuscript by Weijia Lue et al. describes a method to visualise and understand in vivo biodistribution of in situ produced EVs. The manuscript is well written and experimental design is also really good. But due to one technical limitation, results observed in this manuscript cannot be used to draw any conclusions.

A recently published study by Tomoya Hikita in Nature scientific reports "Sensitive and rapid quantification of exosomes by fusing luciferase to exosome marker proteins" shows that the majority of the nanoluc is secreted and is not associated with EVs, even when fused to EV proteins, suggesting that nanoluc cleaves itself in the producer cell and release itself as a free protein. This is the same observation we have made in our own experiments (unpublished). Therefore, the distribution profile observed by the Authors in this manuscript is likely showcasing distribution of free nanoluc protein rather than nanoluc encapsulated in the EVs. This also explain EV accumulation in kidneys, which is not otherwise observed.

Of course, if the authors can prove that all secreted nanoluc is EV associated, the manuscript could be considered for publication.

Reviewer #3 (Remarks to the Author):

The article by Luo et al is innovative which uses a small-sized nano-luciferase-CD63 expressed in the exosomes released specifically by cardiomyocytes (CM) using alpha-MHC tamoxifen inducible promoter. This article also provides a comprehensive characterization of the exosomes released by CM, its presence in plasma and other organs, its effect on cardiomyocyte morphology and function and uptake by different tissues and by fibroblasts in the heart, which is very interesting. The research design is excellent, controls appropriate, writing is succinct, and data are presented clearly. Overall, this new mouse model demonstrates that CMs release exosomes in physiological conditions. Going forward, this will provide an innovative useful tool to the cardiovascular research community to study in vivo release of exosomes specifically from cardiomyocytes.

Specific comments:

1. The exosomes were isolated using ultrafiltration (0.22µm) followed by exoquick. Exoquick is a precipitation method, therefore the authors need to clarify in the result/discussion that there could be vesicle-independent proteins (vesicle-free CD63?), lipids and RNA, which are likely to be precipitated by such isolation methods.
2. In Fig3c, it would be good to include the cardiac/cardiomyocyte levels of luc activity, which will provide a comparative assessment of origin vs. uptake.
3. The authors may need to elaborate the benefits and impact of this mouse model in the discussion in light of existing cardiovascular/cardiomyocyte exosomes literature– that even primary culture of cardiomyocytes contains contaminating fibroblasts/endothelial cells, therefore study of exosomes released by CMs is extremely difficult.

4. The authors need to clarify a minor limitation of the study that due to the known heterogeneity of the exosomes population, CD63 protein may not be present all exosomes secreted by a cell/cells. Therefore, researchers using this mouse model need to be aware of this limitation to design any experiments.

Reviewer #4 (Remarks to the Author):

Summary:

In this manuscript entitled, 'Spatial and temporal control of endogenous exosome tracking in mouse', by Luo et al describe a new cardiac-specific mouse model to detect cardiomyocyte exosome biodistribution in vivo. The authors develop a tamoxifen-inducible aMHC-CD63NanoLuc reporter mouse to track exosome uptake in vitro and in vivo. The improvement of this system relative to other exosomal luciferase reporters is an extended half-life as well as its specificity to cardiomyocytes. The authors employ some in vitro and in vivo techniques to support their claims but require numerous additional experiments prior to consideration for publication.

Major Comments:

- 1) Numerous grammatical and typographical errors make the manuscript difficult to follow. A thorough review of the manuscript text should be performed.
- 2) CD63-NanoLuc constructs, as well as CD9- and CD81-NanoLuc, have been reported (e.g., Hikita, Sci Rep, 2018). These papers need to be cited and incorporated appropriately within this manuscript.
- 3) Title. The title is misleading and does not reflect the focus of the paper. The premise of this paper is the development of a new mouse model to track the biodistribution of cardiomyocyte-specific exosomes.
- 4) Exosomes may be isolated through various techniques. The Exoquick methodology used here causes exosomal clumping and precipitates non-exosomal entities (e.g., secreted proteins). It would be worthwhile to complement these studies with a purer isolation protocol, such as ultracentrifugation or ultrafiltration by centrifugation. Furthermore, it is essential that the author details the method of exosomal isolation within the results sections of the text, not only in the methodology section.
- 5) Figure 1A. This figure should include the dosing protocol for tamoxifen (route of administration, concentration, and frequency).
- 6) Figure 1B. This figure does not have enough detail. What is the size of the transgene? Where is the WT band?
- 7) Figure 2. Should show exosome characterization (i.e., current Figure 5 should be inserted here, and the current Figure 2 should be moved to a new Figure 3).
- 8) Figure 2, A&B. It is unclear why the authors chose to only compare the expression of CD63NanoLuc transcript between heart and skeletal muscle. All other tissues, as shown in Figure 3C, should be incorporated into this figure.
- 9) Figure 2, C&D. Better labeling is required to clarify what TG and R26CreERT reflect. Additionally, a detailed protocol should be incorporated into this figure to show time course and administration of Tamoxifen and NanoLuc substrate.
- 10) Figure 2D. I presume this figure reflects a single administration of NanoLuc substrate. What happens with multiple administrations?
- 11) Figure 2, E&F. This figure should incorporate all major cell types in the heart (i.e., cardiomyocytes, fibroblasts, endothelial cells, and smooth muscle cells) and reflect the methodology used for cell isolation. To demonstrate cell purity, each population should reflect cell-specific transcripts (e.g., aMHC for cardiomyocytes) alongside the expression of the CD63NanoLuc transcript.
- 12) Figure 3B. What time point were plasma exosomes assessed for luciferase activity? A time course should be provided to show the decay over time.

- 13) Figure 3C. Representative bioluminescence images should be inserted for all expression analyses.
- 14) Figure 4. Additional cardiac cell types (described in point 10) should be examined here for exosome uptake. It is important to compare/contrast the uptake of cardiomyocyte exosomes between cardiac cell types.
- 15) Figure 5. As stated earlier, this figure should be presented as Figure 2.
- 16) Figure 5. NTA graphs for all exosome populations should be incorporated.
- 17) Figure 6. The relevance of this figure is questionable. The authors do not state the timepoint for cardiac functional and structural measurements, nor do they provide a baseline measurement (prior to administration of compounds [e.g., tamoxifen or NanoLuc substrate]). The authors must provide baseline measurements and multiple time points following treatment (at least weeks post-treatment).

Minor Comments:

- 1) The authors state several points throughout the manuscript without appropriately citing their references. A thorough review of all cited works should be performed.
- 2) Fluorescent dyes and radioisotope labels for exosomes are not membrane-conjugated entities, as stated in the introduction.

We thank the reviewers and editors for the constructive comments. We have added a significant amount of new data in response to the comments. The following is the summarization pertaining to the reviewer's specific comments point-by-point.

Reviewer #1 (Remarks to the Author):

The manuscript "Spatial and temporal control of endogenous exosome tracking in mouse" by Luo et al. aims to provide a new mouse model for tracking endogenous cardiac exosomes. It builds on the use of NanoLuc that reaches emission 150-fold stronger than the signal emitted by Firefly and Renilla luciferase. Such tool should allow tracking of EVs at the whole-animal scale. Although the aim of this manuscript and research is of great interest to the field, the authors unfortunately fall short in demonstrating that they actually track exosomes.

Major comments:

1. - Specificity of CD63-NanoLuc for the tracking of exosomes: In order to fully prove that organ inter-communication is mediated by exosome-mediated CD63-NanoLuc, the authors could compare the behavior of a similar mouse tagged with NanoLuc only (not tagged to CD63). Besides, CD63 has been fused to its C-terminal region known to contain regulatory domains of CD63 : have the authors assessed whether this impact number and morphology of EVs, both in culture and in mouse blood circulation ?

Response: We appreciate the reviewer to bring this interesting question. To find the answer for this question, we have compared the tissue distribution of free NanoLuc versus CD63NanoLuc labeled exosomes. Free NanoLuc was introduced via tail vein injection. The data exhibited a distinct distribution pattern of the free NanoLuc, in which almost all of the free NanoLuc went to the liver. The obviously different distribution patterns exhibited by NanoLuc and CD63NanoLuc clearly suggested that the CD63NanoLuc faithfully represents the exosome tissue distribution in the mouse model. The new data are presented at Supplement Fig. 2b.

To address the concern if there is any impact on the number and morphology of exosomes due to the fusion of NanoLuc to the C-terminal of CD63, we have assessed the morphology, particle size and numbers in the exosomes isolated from cell cultures and animal plasma as suggested by the reviewer. Two sets of control groups, R26CreERT2 with Tamoxifen and R26CreERT2;TG- α MHC-STOP-CD63NanoLuc without tamoxifen, were applied in order to meet the rigorous conditions. TEM and NTA analysis showed that CD63NanoLuc had no obvious impact on the exosome morphology, particle size and its numbers. The new data were presented in Fig. 3.

Additionally, the uptake of exosomes were evaluated *in vitro* and *in vivo*. The data showed that the fusion of CD63 to NanoLuc had no effect on the exosome uptake (Fig. 5). Our finding is consistent with recent one report showing that NanoLuc knock-in fusion to CD63 C-terminal in human cancer cell has no impact on exosome biogenesis and biological functions¹.

Finally, the long-term effect of CD63NanoLuc on cardiac morphology and function was evaluated (Supplement Fig. 3). No deleterious effect was found, suggesting that NanoLuc fusion did not affect CD63 non-exosome function.

2. - Along this line, purification of EVs is performed by ExoQuick, which is known to be relatively prone to artefacts. This referee would recommend performing classical ultracentrifugation (and gradients) for ensuring quality of the purified and to avoid the potential contamination of non-EVs CD63-LUC protein or mRNA.

Response: We took the reviewer's suggestion and have used ultracentrifugation to purify the exosomes from the culture supernatant of primary cardiomyocytes. The results are presented in Fig. 3d-e.

3. - The manuscript frustratingly lacks a clear demonstration that the tool could be used to track shedding and fate of cardiac exosomes in vivo. In order to validate this approach, one should demonstrate that luciferase activity could be used in whole-animal to track the fate of exosomes shed from the cardiomyocytes. Such information is partly contained in Fig.3c but the benefit of using a brighter form of luciferase to perform imaging in a living animal is not demonstrated.

Response: Like any technology, the NanoLuc tracking mouse model has its advantage and limitation. The NanoLuc is the most sensitive reporter currently available emitting 150-fold stronger luminescent signal compared to the traditional luminescent signal. This study has successfully demonstrated the luminescent image in the live animal heart in vivo. Furthermore, highly sensitive luminescent signals have been quantitatively exhibited and demonstrated in 13 major organs, providing a useful tool for exosome research.

However, we share with the reviewer's point, and wish that the exosomes specifically-released from the animal cardiomyocytes could be imaged in whole animal body by luminescent imaging. We calculated that the cardiac exosomes only contributed to 1.4‰ of the plasma exosomes. Obviously, such a weak signal is beyond the capability for imaging detection, which is the limitation of this mouse model. We have discussed this limitation in the manuscript. However, even with this limitation, this model still enables us to measure and quantify endogenous cardiac exosomes uptaken by the animal organs and even cells, which is the ultimate goal of this study.

4. - The aim of developing a cardiomyocyte-specific transgene isn't clear and not demonstrated in this manuscript – one should demonstrate that exosome released by cardiomyocytes are functional and that such function can be tracked using this new imaging tool.

Response: We took the reviewer's suggestion and have revised the manuscript by elaborating the significance of this animal model for cardiac exosome study. Meanwhile, we would like to indicate that the purpose of animal heart chosen in this study was to demonstrate a spatial control of the endogenous exosome tracking. It is a proof-of-concept animal model. The biological consequences of exosomes released from cardiomyocytes would be interesting for future study, which is not the scope of the current study.

Regarding the function and the tracking of exosomes released from the cardiomyocytes, we have addressed the issues in the response to the comment number #1.

Minor comments:

- 1. To further ensure specificity and exclude promoter leakiness, overall luminescence signal (photon flux) should be quantified in fig.2c to ensure absence of non-specific expression.

Response: We took the reviewer's suggestion and have added the photon flux in Fig. 2c.

The absence of non-specific expression and the non-leakage were demonstrated not only by the luminescent imaging, but also by the quantitative PCR and luciferase assay, the more sensitive methods than the imaging assessment (Fig. 2a&b, Supplement Fig. 1).

- 2. a similar quantification for the signal obtained in the heart, and the entire animal, after substrate injection should be performed to provide a clear description of the transgene biodistribution and lifetime/halflife

- 4. Figure 3c requires additional proof of the specificity of the signal for exosome CD63-NanoLuc : could the authors provide IVIS images or additional proof that exosomes can be found in these organs ? Maybe, providing control images of similar animal/organs in absence of tamoxifen would support the claims. In addition, when the tamoxifen treatment is stopped, how long can the luciferase signal be detected ? Such comparison would allow to assess EVs half-life.

Response: We addressed the comment 2 and 4 together due to the similarity of the two questions.

The bio-distribution of the CD63NanoLuc reporter has been quantified in Fig. 4c in 13 major organs. The precise measurement of "life time/half-life" of the NanoLucCD63 protein was beyond the scope of this study, and cannot be measured by this animal model. The expression of the NanoLucCD63 was turned on and remains constitutively active upon tamoxifen induction. The decay of the luminescent signal along with time was due to the elimination of the substrate, and not because of the degradation of the NanoLuc CD63 protein. The latter was permanently (constitutively) expressed after tamoxifen induction.

The background of animal/organs luminescent signal without tamoxifen treatment has been added in Supplement Fig. 1b as suggested.

- 3. additional controls and detail of exosome purity and presence should be provided for figure 3a and 3b (see major comments)

Response: Yes, two sets of control groups, R26CreERT2 with tamoxifen and R26CreERT2;TG- α MHC-STOP-CD63NanoLuc without tamoxifen, has been applied to meet the rigorous conditions as explained in the response to comment one. The new data are presented in Fig. 3. The detail exosome purification procedures are presented in the online method section.

- 5. it is not clear, in Fig.5a, where and how were the exosomes purified. The authors should be more clear, along the manuscript, and carefully detail what has been done.

Response: We took the reviewers suggestion and added more details in the method.

- data provided in figure 6 are useful but are only control experiments demonstrating that overexpression of this construct does not affect cardiac development. As such, this should be a supplemental information.

Response: We took the reviewers suggestion and have moved Fig. 6 to Supplement Fig. 3.

- the manuscript would benefit from english proofreading

Response: We have improved English by prove reading.

- the introduction is rather superficial considering the amount of approaches that have been already published (reviewed in Hyenne et al. CAM 16) and lacks referencing to two recent methods that have been published, in particular using the zebrafish embryo (although one of them is mentioned superficially in the discussion).

Response: Yes, we have cited the suggested review article and discussed the current genetic animal models including the zebrafish model.

- in the discussion, when the authors compare injection of exogenous EVs to endogenous, they should discuss the entry/injection routes.

Response: Yes, we have added the discussion about the different entry/injection routes.

1. Hikita, T., Miyata, M., Watanabe, R. & Oneyama, C. Sensitive and rapid quantification of exosomes by fusing luciferase to exosome marker proteins. *Sci Rep* **8**, 14035 (2018).

Reviewer #2 (Remarks to the Author):

Manuscript by Weijia Lue et al. describes a method to visualise and understand in vivo biodistribution of in situ produced EVs. The manuscript is well written and experimental design is also really good. But due to one technical limitation, results observed in this manuscript cannot be used to draw any conclusions. A recently published study by Tomoya Hikita in Nature scientific reports "Sensitive and rapid quantification of exosomes by fusing luciferase to exosome marker proteins" shows that the majority of the nanoluc is secreted and is not associated with EVs, even when fused to EV proteins, suggesting that nanoluc cleaves itself in the producer cell and release itself as a free protein. This is the same observation we have made in our own experiments (unpublished). Therefore, the distribution profile observed by the Authors in this manuscript is likely showcasing distribution of free nanoluc protein rather than nanoluc encapsulated in the EVs. This also explain EV accumulation in kidenys, which is not otherwise observed.

Of course, if the authors can prove that all secreted nanoluc is EV associated, the manuscript could be considered for publication.

Response: We thank the reviewer to bring this interesting and important issue. We carefully went through the Scientific reports paper by Tomoya Hikita¹. Here is what we found.

The authors observed that "Ultracentrifugation drastically reduced the intensity of luminescence in the culture medium" (Fig. 1c and supplement Fig.5). The authors interpreted the observations by "This implies that most of the exosomes in samples are lost during the ultracentrifugation or washing process".

Therefore, we could not find any evidence from this paper to support the reviewer's comments, in which "...the majority of the nanoluc is secreted and is not associated with EVs, even when fused to EV proteins... suggesting that nanoluc cleaves itself in the producer cell and release itself as a free protein".

To elucidate and distinguish the distribution of free NanoLuc protein, we introduced 1×10^7 RLU (relative luciferase unit) of free NanoLuc protein by tail vein injection, which was comparable to the total RLU in the transgenic mouse plasma. We compared the tissue distribution of the free NanoLuc versus exosome-associated CD63NanoLuc. The data exhibited a distinct distribution pattern of the free NanoLuc, in which almost all of the free NanoLuc went to the liver. These obviously different distribution patterns clearly suggested that the CD63NanoLuc faithfully represents the endogenous exosome tissue distribution in our mouse model. The new data are presented in Supplement Fig. 2b.

Finally, we measured the percentage of the exosome-free NanoLuc over the total luminescent intensity in the plasma. We found that the exosome-free NanoLuc contributed to 7% of the total luciferase activity in the plasma (Supplement Fig. 2c), suggesting that the luciferase tissue distribution mainly came from CD63NanoLuc-labeled exosomes.

1. Hikita, T., Miyata, M., Watanabe, R. & Oneyama, C. Sensitive and rapid quantification of exosomes by fusing luciferase to exosome marker proteins. *Sci Rep* **8**, 14035 (2018).

Reviewer #3 (Remarks to the Author):

The article by Luo et al is innovative which uses a small-sized nano-luciferase-CD63 expressed in the exosomes released specifically by cardiomyocytes (CM) using alpha-MHC tamoxifen inducible promoter. This article also provides a comprehensive characterization of the exosomes released by CM, its presence in plasma and other organs, its effect on cardiomyocyte morphology and function and uptake by different tissues and by fibroblasts in the heart, which is very interesting. The research design is excellent, controls appropriate, writing is succinct, and data are presented clearly. Overall, this new mouse model demonstrates that CMs release exosomes in physiological conditions. Going forward, this will provide an innovative useful tool to the cardiovascular research community to study in vivo release of exosomes specifically from cardiomyocytes.

Specific comments:

1. The exosomes were isolated using ultrafiltration(0.22um) followed by exoquick. Exoquick is a precipitation method, therefore the authors need to clarify in the result/discussion that there could be vesicle-independent proteins (vesicle-free CD63?), lipids and RNA, which are likely to be precipitated by such isolation methods.

Response: We thank the reviewer's constructive suggestion and have applied the ultracentrifugation, a more commonly used method, to the purification of exosomes in several experiments. The new data were presented in Fig. 3 d-e.

Because the luciferase assay is the major assessment for the isolated exosomes in this mouse model, the impact of lipids and RNAs on the result is expected to be minimal. We have also measured the percentage of the exosome-free NanoLuc over the total NanoLuc by luciferase activity in the plasma. We found that the exosome-free NanoLuc contributed to only 7% of the total luciferase activity (Supplement Fig. 2c), which is the highest possible vesicle-free NanoLuc protein contaminations that impact the exosome assessments.

2. In Fig3c, it would be good to include the cardiac/cardiomyocyte levels of luc activity, which will provide a comparative assessment of origin vs. uptake.

Response: We thank the reviews suggestions and have compared Fig. 4c with Fig. 2b & f in the discussion.

3. The authors may need to elaborate the benefits and impact of this mouse model in the discussion in light of existing cardiovascular/cardiomyocyte exosomes literature– that even primary culture of cardiomyocytes contains contaminating fibroblasts/endothelial cells, therefore study of exosomes released by CMs is extremely difficult.

Response: We thank the reviewer for the suggestions and have revised the manuscript. We discussed the benefits and impacts of this mouse model in light of existing cardiovascular/cardiomyocyte exosomes literatures.

4. The authors need to clarify a minor limitation of the study that due to the known heterogeneity of the exosomes population, CD63 protein may not be present all exosomes secreted by a cell/cells. Therefore, researchers using this mouse model need to be aware of this limitation to design any experiments.

Response: That is a very good suggestion and we have discussed the issues, limitations et al. in the discussion section.

Reviewer #4 (Remarks to the Author):

Summary:

In this manuscript entitled, 'Spatial and temporal control of endogenous exosome tracking in mouse', by Luo et al describe a new cardiac-specific mouse model to detect cardiomyocyte exosome biodistribution in vivo. The authors develop a tamoxifen-inducible aMHC-CD63NanoLuc reporter mouse to track exosome uptake in vitro and in vivo. The improvement of this system relative to other exosomal luciferase reporters is an extended half-life as well as its specificity to cardiomyocytes. The authors employ some in vitro and in vivo techniques to support their claims but require numerous additional experiments prior to consideration for publication.

Major Comments:

1) Numerous grammatical and typographical errors make the manuscript difficult to follow. A thorough review of the manuscript text should be performed.

Response: We took the reviewer's suggestion and had a proof reading for the manuscript.

2) CD63-NanoLuc constructs, as well as CD9- and CD81-NanoLuc, have been reported (e.g., Hikita, Sci Rep, 2018). These papers need to be cited and incorporated appropriately within this manuscript.

Response: We have cited and discussed the reference as suggested by the reviewer.

3) Title. The title is misleading and does not reflect the focus of the paper. The premise of this paper is the development of a new mouse model to track the biodistribution of cardiomyocyte-specific exosomes.

Response: We chose the animal heart to demonstrate that the tracking system can be controlled spatially. This is a proof-of-concept mouse model. While it is particularly valuable for the exosome study in cardiovascular research, the model has a general significance for a broad spectrum of audiences in scientific community. The title represents this point and fits the scope of the study.

4) Exosomes may be isolated through various techniques. The Exoquick methodology used here causes exosomal clumping and precipitates non-exosomal entities (e.g., secreted proteins). It would be worthwhile to complement these studies with a purer isolation protocol, such as ultracentrifugation or ultrafiltration by centrifugation. Furthermore, it is essential that the author details the method of exosomal isolation within the results sections of the text, not only in the methodology section.

Response: We have taken the reviewer's suggestion and used ultracentrifugation to process the purification of exosomes. More details of the method have been added to the text.

5) Figure 1A. This figure should include the dosing protocol for tamoxifen (route of administration, concentration, and frequency).

Response: We have updated Fig. 1c with the addition of the suggested information.

6) Figure 1B. This figure does not have enough detail. What is the size of the transgene? Where is the WT band?

Response: We have labeled the DNA ladder indicating the size of the genotyping product for the transgene assessment in Fig. 1b. Since the genotyping PCR only detects the transgene, there should be no wild-type (WT) band. Only the animal DNA carries the transgene will give a band in the genotyping assessment.

7) Figure 2. Should show exosome characterization (i.e., current Figure 5 should be inserted here, and the current Figure 2 should be moved to a new Figure 3).

15) Figure 5. As stated earlier, this figure should be presented as Figure 2.

Response: We address the comment 7 and 15 together due to the similarity of the two questions.

We took the reviewer's suggestion to re-arrange the two figures. Since we have added a set of new data, the updated data are presented in fig. 3.

8) Figure 2, A&B. It is unclear why the authors chose to only compare the expression of CD63NanoLuc transcript between heart and skeletal muscle. All other tissues, as shown in Figure 3C, should be incorporated into this figure.

Response: We have assessed the transcript and luciferase activity in the rest of tissues as suggested by the review. The negative control results from these tissues are added in Supplement Fig.1.

9) Figure 2, C&D. Better labeling is required to clarify what TG and R26CreERT2 reflect. Additionally, a detailed protocol should be incorporated into this figure to show time course and administration of Tamoxifen and NanoLuc substrate.

Response: The representations of the abbreviations in the figures have been added to the figure legends as suggested by the reviewer.

10) Figure 2D. I presume this figure reflects a single administration of NanoLuc substrate. What happens with multiple administrations?

Response: A continual luminescent signal is expected with multiple administrations of substrate.

11) Figure 2, E&F. This figure should incorporate all major cell types in the heart (i.e., cardiomyocytes, fibroblasts, endothelial cells, and smooth muscle cells) and reflect the methodology used for cell isolation. To demonstrate cell purity, each population should reflect cell-specific transcripts (e.g., aMHC for cardiomyocytes) alongside the expression of the CD63NanoLuc transcript.

Response: The tissue specificity of transgene expression driven by aMHC promoter has been demonstrated by many studies for more than two decades. We chose cardiac fibroblasts, consist of 90% of the non-cardiomyocytes in the heart, to confirm this specificity.

12) Figure 3B. What time point were plasma exosomes assessed for luciferase activity? A time course should be provided to show the decay over time.

Response: The plasma exosomes were assessed one week after tamoxifen induction.

The expression of reporter transgene CD63NanoLuc is turned on and remains constitutively active upon tamoxifen induction. The luminescent signal will continuously be detectable as long as the substrate is

available. The decay of the luminescent signal along with time is due to the clearance of the substrate in the body.

13) Figure 3C. Representative bioluminescence images should be inserted for all expression analyses.

Response: We share with the reviewer's point, and wish that the exosomes specifically-released from the animal cardiomyocytes could be imaged in whole animal body by luminescent imaging. We calculated that the cardiac exosomes only contribute to 1.4‰ of the plasma exosomes. Obviously, such a weak signal is beyond the capability for imaging detection, which is the limitation of this mouse model. However, even with this limitation, this model still enables the detection and quantification of cardiac exosomes uptaken by the animal organs and even cells through luciferase activity assessment, providing the valuable tissue distribution profile for endogenous exosomes, which meets the ultimate goal of this study.

14) Figure 4. Additional cardiac cell types (described in point 10) should be examined here for exosome uptake. It is important to compare/contrast the uptake of cardiomyocyte exosomes between cardiac cell types.

Response: The main purpose of Fig. 4 is to demonstrate exosome-mediated intercellular communication *in vivo*. Cardiac fibroblasts, consist of 90% of the non-cardiomyocytes in the heart, are used for the prove-of-concept purpose. This model provided the evidence to show the paracrine effect mediated by exosome in mice for the first time.

16) Figure 5. NTA graphs for all exosome populations should be incorporated.

Response: We have added additional NTA graphs as recommended (Fig 3b & e).

17) Figure 6. The relevance of this figure is questionable. The authors do not state the time point for cardiac functional and structural measurements, nor do they provide a baseline measurement (prior to administration of compounds [e.g., tamoxifen or NanoLuc substrate]). The authors must provide baseline measurements and multiple time points following treatment (at least weeks post-treatment).

Response: We thank the reviewer's comment, and have added the time point for the cardiac functional assessment, which is 3 month after tamoxifen induction.

Equally important, we have practiced the scientific rigor principle and included five control groups in the animal cardiac functional assessments to achieve the highest standard and stringency. We believe that these five control groups have covered the conditions that could potentially affect the outcome of the study.

Minor Comments:

1) The authors state several points throughout the manuscript without appropriately citing their references. A thorough review of all cited works should be performed.

Response: We have revised the manuscript coordinately.

2) Fluorescent dyes and radioisotope labels for exosomes are not membrane-conjugated entities, as stated in the introduction.

Response: We appreciate the suggestion and have revised the paper coordinately.

Reviewers' comments:

Reviewer #1 (Remarks to the Author):

We appreciate the efforts of the authors to address the different points that we raised in our initial review. Although most of the new data presented are satisfying, we still have a few points that we think should be addressed.

1. Specificity of the CD63-nanoLuc for the tracking of exosomes.

The authors use two arguments to prove the specificity of the signal observed: i) injection of free nanoluc in the mouse circulation quickly vanishes and accumulates in the liver, which is different from the organ accumulation observed in CD63-nanoluc transgenic mice. ii) Separation of mouse plasma (as we understand coming from CD63-nanoLuc transgenic mice, but this should be explicitly written in the figure legend and in the main text) based on molecular weight, revealing that 7% of the luciferase signal corresponds to entities below 100kDa.

In the first point i), different things are compared (intravenous injection vs natural release), which could impact the conclusions. While we understand that generating a transgenic mouse line expressing free nanoLuc might be too complex, this question could be resolved by comparing intravenous injection of soluble nanoLuc to intravenous injection of CD63-nanoLuc EVs properly isolated (by ultracentrifugation).

Regarding the second point ii), while the experiment is already informative as it reveals the presence of a minority population of reporter not associated with EVs, it would have been more convincing to properly isolate EVs from the plasma (by ultracentrifugation) and compare the luciferase activity of the EV fraction with the non-EV fraction.

In the main text, the fact that at least 7% of the total signal does not correspond to EV bound CD63 should be acknowledged.

2. Impact of CD63-nanoluc overexpression to EV secretion number and morphology. Again, we appreciate the efforts of the authors, but we still have a few concerns. The electron microscopy images in panel A are unlikely to be EVs, by contrast to the ones presented in panel D where cup shape vesicles are clearly visible. This is likely to be linked to the method of isolation chosen. A is exoquick while D is ultracentrifugation. Ideally ultracentrifugation should be performed from mouse plasma. Alternatively, more convincing images should be presented for A. In panel C, why are there two lanes per condition? In panel B, the size distributions seem to be similar in transgenic mice with or without tamoxifen, but different from the R26CreERT2 mice. This should simply be notified.

In their rebuttal authors say that the nanoLuc fusion does not affect exosome uptake. No evidence is presented to support this claim, since no precise comparison of uptake was done between EVs +/- nanoLuc.

3. Isolation of EVs. We are convinced by the data presented in figure 3d and e. However the panels A-C show material isolated with exoquick. As mention earlier, it would be important to properly isolate EVs from the plasma (by ultracentrifugation) and compare the luciferase activity of the EV fraction with the non-EV fraction.

Reviewer #4 (Remarks to the Author):

In this revised paper, the authors provide additional information on a new cardiac-specific mouse model to detect cardiomyocyte exosome biodistribution *in vivo*. The first version of the paper lacked significant amounts of confirmatory data and, in the revised version, many of the same weaknesses exist. Several of the original comments were inadequately addressed and the authors do not provide much new data to clarify the biodistribution of cardiomyocyte-specific exosomes *in vivo*.

Comments:

- 1) Grammatical errors still exist throughout the manuscript.
- 2) The authors do not clearly indicate where in the text where revisions were performed, making the review process challenging.
- 3) The authors state that this manuscript is a proof-of-concept mouse model, but in fact it is a cardiomyocyte-specific exosome biodistribution paper. The authors must change their title to reflect the data within the manuscript, as previously stated in my original set of comments.
- 4) The authors mention the use of ultracentrifugation as an exosome isolation method, but do not clearly show which data sets and figures incorporated the ultracentrifugation isolation procedure. The data presented suggest that ultracentrifugation was only used in Figure 3, but not throughout the rest of the manuscript.
- 5) A comparison is performed between cardiomyocytes and cardiac fibroblasts is performed to demonstrate specificity in uptake of cardiomyocyte-derived exosomes. As mentioned in the last set of comments, other major cardiac cell types need to be explored. While the authors state that 90% of the heart is comprised of fibroblasts (in response to Reviewer 4, comments 11 and 14), this statement is inaccurate. Numerically, cardiomyocytes comprise ~30% of all cells in the myocardium and endothelial cells, which are the most abundant cell type, comprise >50% of all cells in the myocardium. The authors should refer to Pinto AR, *Circ Res*, 2016 for accurate quantitative cell measurements.
- 6) The biodistribution revealed by NanoLuc substrate infusion suggests that the signal reflects endosomal exosomes prior to extracellular release. Without any images to support the release and uptake of exosomes within isolated cells (cardiomyocytes, fibroblasts, and endothelial cells) of the myocardium the conclusions are speculative at best. Additionally, the authors should quantitate by NTA the number of exosomes released by cardiomyocytes in culture. This would provide the resolution to determine if the number of exosomes released by cardiomyocytes into circulation is within the limit of detection; as the authors state in their response to reviewers, the luminescence signal is extremely low in circulation (~1.4%) and undetectable by current techniques.
- 7) As mentioned in my previous comments, administration of NanoLuc substrate at multiple timepoints should be explored (Reviewer 4, comment 10) as it would provide improved resolution of the systemic biodistribution of exosomes. This comment was inadequately addressed in this revision.

Point-by-point response to reviewer #1:

We appreciate the efforts of the authors to address the different points that we raised in our initial review. Although most of the new data presented are satisfying, we still have a few points that we think should be addressed.

Q: 1. Specificity of the CD63-nanoLuc for the tracking of exosomes.

The authors use two arguments to prove the specificity of the signal observed: i) injection of free nanoluc in the mouse circulation quickly vanishes and accumulates in the liver, which is different from the organ accumulation observed in CD63-nanoluc transgenic mice. ii) Separation of mouse plasma (as we understand coming from CD63-nanoLuc transgenic mice, but this should be explicitly written in the figure legend and in the main text) based on molecular weight, revealing that 7% of the luciferase signal corresponds to entities below 100kDa.

In the first point i), different things are compared (intravenous injection vs natural release), which could impact the conclusions. While we understand that generating a transgenic mouse line expressing free nanoLuc might be too complex, this question could be resolved by comparing intravenous injection of soluble nanoLuc to intravenous injection of CD63-nanoLuc EVs properly isolated (by ultracentrifugation).

R: This is the same concern repeated by the reviewer, which is whether the tissue distribution of free NanoLuc molecules contributes significantly to the tissue distribution of CD63-NanoLuc. In other words, whether these two particles exhibit or behavior similarly in their tissue distributions.

Before we address the question, we would like to point out two discoveries from this study.

First, cardiomyocyte-released exosomes only comprise 1.4‰ (0.14%) of the plasma exosomes; Secondly, 93% of luciferase activity in plasma is due to exosome-associated CD63NanoLuc and 7% was due to exosome-free NanoLuc.

Clearly, the proportion of free NanoLuc in mouse plasma is minimal. The answer is obvious: the contribution of free NanoLuc molecules to the tissue distribution is minimal. The CD63-NanoLuc reporter faithfully represents cardiac exosomes *in vivo*.

Even with the clear evidence, we respect the reviewer and follow reviewer's suggestion and has compared the tissue distribution of NanoLuc molecules versus CD63NanoLuc-labelled exosomes in our previous revision. As the reviewer indicated that "*generating a transgenic mouse line expressing free nanoLuc might be too complex*", we introduced free NanoLuc via tail vein injection. The data exhibited a distinct distribution pattern of the free NanoLuc compared to CD63NanoLuc-labelled exosomes. Almost all of the free NanoLuc went to the liver.

The obviously different distribution pattern of the two particles once again demonstrates the specificity of CD63NanoLuc, which faithfully represents the exosome tissue distribution. The data were presented in Supplement Fig. 2b in the previous revision.

However, to our surprise in this 2nd revision, the reviewer further insisted "*comparing intravenous injection of soluble NanoLuc to intravenous injection of CD63-nanoLuc EVs properly isolated (by ultracentrifugation)*". How can the injection of two additional exogenous particles further demonstrate the specificity of the endogenous CD63-NanoLuc reporter?

The key advantage of this study is to enable tracking endogenous exosomes in mice, which is the critical need in the current exosome research. The suggestion to compare two exogenous particles will neither improve the quality nor increase the novelty of the manuscript. There are many studies that have demonstrated or defined the exogenous exosomes introduction in animals.

Q: Regarding the second point ii), while the experiment is already informative as it reveals the presence of a minority population of reporter not associated with EVs, it would have been more convincing to properly isolate EVs from the plasma (by ultracentrifugation) and compare the luciferase activity of the EV fraction with the non-EV fraction

R: We disagree with the reviewer.

The usage of ultracentrifugation will not deliver a “*more convincing*” result. Oppositely, one recent study found that the retrieval rate of exosomes by ultracentrifugation was about 20%, suggesting a significant loss of exosomes during the ultracentrifugation or washing procedures¹.

We used 100kDa protein concentrator to separate the EV-labeled with Nanoluc (>200kDa) from free Nanoluc (19kDa) and free CD63Nanoluc (45kDa), which has minimized the loss of exosomes and provided a much more accurate assessment for each population than the ultracentrifugation suggested by the reviewer.

Q: 2. Impact of CD63-nanoluc overexpression to EV secretion number and morphology. Again, we appreciate the efforts of the authors, but we still have a few concerns. The electron microscopy images in panel A are unlikely to be EVs, by contrast to the ones presented in panel D where cup shape vesicles are clearly visible. This is likely to be linked to the method of isolation chosen. A is Exoquick while D is ultracentrifugation. Ideally ultracentrifugation should be performed from mouse plasma. Alternatively, more convincing images should be presented for A.

R: We disagree with the reviewer’s assumption.

The heterogeneity of exosomes is widely recognized². We have included two TEM images of exosomes cited from respectful publications to show the similarity of theirs compared to ours.

More specifically, the reviewer questioned the images in Fig. 3A and suggested “*Ideally ultracentrifugation should be performed from mouse plasma*”. The suggested experiment will NOT answer the question whether NanoLuc labeling affects exosome morphology. Because cardiomyocyte-released exosomes only comprise 1.4‰ (0.14%) of the plasma exosomes and 99.86% of the plasma exosomes are not labelled, wild-type exosomes. In other words, there is 0.14% cubed chance (2.7×10^{-9}) to catch one exosome released from cardiomyocytes for the comparison of TEM images among three groups.

The appropriate approach is to collect exosomes directly from isolated cardiomyocytes, which ensures that those exosomes come from the cardiomyocytes. We have used ultracentrifugation for the isolation of these cardiac exosomes in our previous revision to respect the reviewer’s suggestion. Both TEM imaging and NTA results were presented in Fig. 3D and 3E, respectively.

J Clin Invest. 2016;126(4):1216-1223. <https://doi.org/10.1172/JCI81136>. Fig. 1.

<https://www.lsbio.com/assaykits/exosome-isolation-kit-serum-plasma-ls-k785/785>

Q: In panel C, why are there two lanes per condition?

R: We respectfully express our surprise by the limitation of the reviewer's knowledge in mouse animal study.

In Fig. 3 panel C, the two lanes represent the data from two individual animals. It is a common knowledge to include biological repeats, which is a standard principle of datum presentation in order to keep experimental stringency.

Q: In panel B, the size distributions seem to be similar in transgenic mice with or without tamoxifen, but different from the R26CreERT2 mice. This should simply be notified.

R: We disagree with the reviewer and would like to cite one example to support our result about the size distribution.

In this elegant study, PHluorin (27kDa, bigger than the 19kDa NanoLuc) as a tag was fused to the C-terminal of CD63, which is similar to our strategy. The author compared the size distribution of labeled- versus wild-type exosomes. The minor differences between the two groups were exhibited and highlighted by rectangle frame. However, the study concluded that the PHluorin labeling did not affect the exosome size distribution³.

D

Consistently, one recent study using the same NanoLuc-CD63 C-terminus fusion strategy to label exosomes in human cancer cells led to the same result, which the NanoLuc fusion to CD63 C-terminus does not affect exosome biogenesis and functions¹.

Q: In their rebuttal authors say that the nanoLuc fusion does not affect exosome uptake. No evidence is presented to support this claim, since no precise comparison of uptake was done between EVs +/- nanoLuc.

R: We disagree with the reviewer.

The study has provided clear evidence to show that CD63-NanoLuc label exosomes can be up-taken *in vitro* (Fig. 5a) and *in vivo* (Fig. 5b).

We would respectfully ask how to compare any EVs released from specific cells *in vivo* if they are not labeled, as suggested by the reviewer: the comparison of EVs-NanoLuc versus EVs without NanoLuc or any labeling.

How could this comparison constructively improve the study?

Q: 3. Isolation of EVs. We are convinced by the data presented in figure 3d and e. However the panels A-C show material isolated with Exoquick. As mention earlier, it would be important to properly isolate EVs from the plasma (by ultracentrifugation) and compare the luciferase activity of the EV fraction with the non-EV fraction.

R: This is the repeated question raised at the concern 1 and 2.

We would like to express our disappointment and surprise by the recommendations. The reviewer's insistence is out of line with the current understanding/knowledge about EVs isolation methodology.

The reviewer repeatedly insisted the usage of centrifugation method for exosome isolation in all experiments.

In our previous revision, we have respectfully taken the reviewer's suggestion and applied centrifugation in some but not all of experiments. The results are consistent among different isolation methods. We have also explained why the comprehensive approaches are more appropriate and satisfy various situations and have achieved the best results than simply using centrifugation.

Current understanding is that isolation methodology of EVs is in an exploratory stage without a conclusive consensus. There is no single optimal separation method including centrifugation according to the latest Guidelines of International Society for Extracellular Vesicles (ISEV). The guideline has been reiterated by Dr. Kenneth Witwer, the Secretary General and Executive Chair of Science and Meetings of ISEV in the American Heart Association Scientific Sessions 2019 on November 17.

We have conducted careful and exhausted comparisons during this study. Our comprehensive isolation strategy supports the guideline. We cite the paragraph from the guideline as your reference.

"There is no single optimal separation method, so choose based on the downstream applications and scientific question. Separation of non-vesicular entities from EVs is not fully achieved by common EV isolation protocols, including centrifugation protocols or commercial kits that claim EV or "exosome" purification"⁴.

References

1. Hikita, T., Miyata, M., Watanabe, R. & Oneyama, C. Sensitive and rapid quantification of exosomes by fusing luciferase to exosome marker proteins. *Sci Rep* 8, 14035 (2018).
2. Zhang, H. et al. Identification of distinct nanoparticles and subsets of extracellular vesicles by asymmetric flow field-flow fractionation. *Nat Cell Biol* 20, 332-343 (2018).
3. Verweij, F.J. et al. Live Tracking of Inter-organ Communication by Endogenous Exosomes In Vivo. *Dev Cell* 48, 573-589 e574 (2019).
4. Minimal information for studies of extracellular vesicles 2018 (MISEV2018): a position statement of the International Society for Extracellular Vesicles and update of the MISEV2014 guidelines" *J Extracell Vesicles* 7, 1535750 (2018).

Point-by-point response to reviewer #4:

In this revised paper, the authors provide additional information on a new cardiac-specific mouse model to detect cardiomyocyte exosome biodistribution *in vivo*. The first version of the paper lacked significant amounts of confirmatory data and, in the revised version, many of the same weaknesses exist. Several of the original comments were inadequately addressed and the authors do not provide much new data to clarify the biodistribution of cardiomyocyte-specific exosomes *in vivo*.

Comments:

Q: 1) Grammatical errors still exist throughout the manuscript. 2) The authors do not clearly indicate where in the text where revisions were performed, making the review process challenging.

R: The version with tracking change is submitted per the inquiry.

Q: 3) The authors state that this manuscript is a proof-of-concept mouse model, but in fact it is a cardiomyocyte-specific exosome biodistribution paper. The authors must change their title to reflect the data within the manuscript, as previously stated in my original set of comments.

R: We strongly disagree the statement saying that this is a "*cardiomyocyte-specific exosome biodistribution paper*".

We would like to know how the two issues are against each other, "a proof-of-concept mouse model" and "a cardiomyocyte-specific exosome biodistribution paper". Can "a cardiomyocyte-specific exosome biodistribution paper" not have "a proof-of-concept mouse model" or vice versa?

The study has demonstrated the establishment of an animal model for endogenous exosome tracking. We have provided the following evidence to demonstrate that the mice can be used for tracking endogenous exosomes in a temporally and spatially controlled manner and the heart is used as the organ to proof the concept:

- 1) Exosome marker CD63-Nanoluc expression is tightly controlled (Fig 2).
- 2) CD63-Nanoluc expression does not affect exosome secretion, morphology and uptake (Fig 3 & 5a).
- 3) Cardiac exosomes can be labeled by CD63-Nanoluc (Fig. 4a), and these exosomes are released to the blood (Fig. 4b) and can be quantified in the blood.
- 4) By assessing luciferase activities in 13 different organs as examples, we have shown that the inter-organelle exosome uptake can be detected and quantified (Fig. 4C).
- 5) By assessing luciferase activities in fibroblasts as examples, we have demonstrated that the intercellular exosome paracrine effects can be detected and quantified.
- 6) Finally, the study has suggested that long-term expression of CD63-Nanoluc does not result in deleterious effects in the animal.

We are politely asking the reviewer to specify where the paper “*lacked significant amounts of confirmatory data and, in the revised version, many of the same weaknesses exist*” so that we could improve the manuscript.

Q: 4) The authors mention the use of ultracentrifugation as an exosome isolation method, but do not clearly show which data sets and figures incorporated the ultracentrifugation isolation procedure. The data presented suggest that ultracentrifugation was only used in Figure 3, but not throughout the rest of the manuscript.

R: We would like to express our disappointment and surprise by the recommendations. The reviewer’s insistence is out of line with the current understanding/knowledge about EVs isolation methodology.

The reviewer repeatedly insisted the usage of centrifugation method for exosome isolation in all experiments.

In our previous revision, we have respectfully taken the reviewer’s suggestion and applied centrifugation in some but not all of experiments. The results are consistent among different isolation methods. We have also explained why the comprehensive approaches are more appropriate and satisfy various situations and have achieved the best results than simply using centrifugation.

Current understanding is that isolation methodology of EVs is in an exploratory stage without a conclusive consensus. There is no single optimal separation method including centrifugation according to the latest Guidelines of International Society for Extracellular Vesicles (ISEV). The guideline has been reiterated Dr. Kenneth Witwer, the Secretary General and Executive Chair of Science and Meetings of ISEV in the American Heart Association Scientific Sessions 2019 on December 17.

We have conducted careful and exhausted comparisons during this study. Our comprehensive isolation strategy supports the guideline. We cite the paragraph from the Guidelines as your reference.

“There is no single optimal separation method, so choose based on the downstream applications and scientific question. Separation of non-vesicular entities from EVs is not fully achieved by common EV isolation protocols, including centrifugation protocols or commercial kits that claim EV or “exosome” purification”¹.

Q: 5) A comparison is performed between cardiomyocytes and cardiac fibroblasts is performed to demonstrate specificity in uptake of cardiomyocyte-derived exosomes. As mentioned in the last set of comments, other major cardiac cell types need to be explored. While the authors state that 90% of the heart is comprised of fibroblasts (in response to Reviewer 4, comments 11 and 14), this statement is inaccurate. Numerically, cardiomyocytes comprise ~30% of all cells in the myocardium and endothelial cells, which are the most abundant cell type, comprise >50% of all cells in the myocardium. The authors should refer to Pinto AR, Circ Res, 2016 for accurate quantitative cell measurements.

R: The citation is not correct by the reviewer. We would like to present our two previous responses here.

In the “in response to Reviewer 4, comments 11”, here is what we wrote: “We chose cardiac fibroblasts, consist of 90% of **non-cardiomyocytes** in the heart, to confirm this specificity.”

In the comment 14, we wrote again: “Cardiac fibroblasts, consist of 90% of the **non-cardiomyocytes** in the heart, are used for the prove-of-concept purpose.”

The differences are obvious without further clarification.

Q: 6) The biodistribution revealed by NanoLuc substrate infusion suggests that the signal reflects endosomal exosomes prior to extracellular release. Without any images to support the release and uptake of exosomes within isolated cells (cardiomyocytes, fibroblasts, and endothelial cells) of the myocardium the conclusions are speculative at best. Additionally, the authors should quantitate by NTA the number of exosomes released by cardiomyocytes in culture. This would provide the resolution to determine if the number of exosomes released by cardiomyocytes into circulation is within the limit of detection; as the authors state in their response to reviewers, the luminescence signal is extremely low in circulation (~1.4%) and undetectable by current techniques.

R: It is common knowledge that luciferase assay is much more sensitive than luminescent imaging due to the enzymatic amplification of the signal.

By using luciferase assay, we have provided evidence that the CD63-NanoLuc labeled cardiac exosome were released from cardiomyocyte (Fig. 4a) and can be detected in the mouse plasma (Fig. 4b). We have also demonstrated the uptake of CD63-NanoLuc labeled cardiac exosome by fibroblast *in vitro* (Fig. 5A) and *in vivo* (Fig. 5b).

We do not understand why the data from a sensitive and quantitative luciferase activity assay are “speculative”. What is the rationale that the result from a much more sensitive luciferase assay needs to be validated by a much less sensitive luminescent imaging analysis?

The recommendation is deficient in basic scientific knowledge, which is concerned.

The NTA analysis of exosomes released by cardiomyocytes has been presented in Fig. 3E. However, NTA quantification will NOT be able to achieve what the reviewer suggested: “*provide the resolution to determine if the number of exosomes released by cardiomyocytes into circulation is within the limit of detection*”.

The NTA analysis of exosome number is NOT accurate for calculating the total exosome amount since a significant portion of exosomes will be lost during the exosome isolation processes.

Our calculation of the percentage of cardiac exosome in plasma is based on the following formula:

$$\frac{\text{RLU}/\mu\text{g plasma exosomes}}{\text{RLU}/\mu\text{g CM exosomes}}$$

The method is based on exosomes density at 1.15-1.19g/ml by sucrose gradients regardless of the origins, which is widely recognized as natural physics of exosomes and is more accurate².

Q: 7) As mentioned in my previous comments, administration of NanoLuc substrate at multiple timepoints should be explored (Reviewer 4, comment 10) as it would provide improved resolution of the systemic biodistribution of exosomes. This comment was inadequately addressed in this revision.

R: Once again, we are concerned about the recommendation, which is deficient in basic scientific knowledge.

We respectfully responded the reviewer's previous comment 10 and hoped the reviewer would understand the working mechanism of transgene luciferase. Unfortunately, the reviewer has not understood it yet.

Administration of NanoLuc substrate at multiple time points will NOT "*provide improved resolution of the systemic biodistribution of exosomes*". Because the amount of introduced substrates always exceed saturated level in luciferase assays. Therefore, the signal resolution cannot be further improved by "*administration of NanoLuc substrate at multiple timepoints*".

Another example of recommendation with the deficiency in basic understanding of transgenic animal study is the requirement to show the wild-type band in animal genotyping assessment (previous concern #6). We politely pointed out that there should be no wild-type (WT) band in the genotyping assessment (Fig. 1b). Only the transgene will give a band in the genotyping analysis.

References

1. Minimal information for studies of extracellular vesicles 2018 (MISEV2018): a position statement of the International Society for Extracellular Vesicles and update of the MISEV2014 guidelines" *J Extracell Vesicles* **7**, 1535750 (2018).
2. They, C., Amigorena, S., Raposo, G. & Clayton, A. Isolation and characterization of exosomes from cell culture supernatants and biological fluids. *Curr Protoc Cell Biol* Chapter 3, Unit 3 22 (2006).

REVIEWERS' COMMENTS:

Reviewer #1 (Remarks to the Author):

To the authors:

Since the authors refused all suggestions, even the simplest text changes, we will not continue endless discussions but just stick to two important points:

1. Specificity of the CD63-nanoLuc for the tracking of exosomes.

The central point is the demonstration of the specificity of the signal. The separation of mouse plasma by molecular weight does not provide proofs that part of the signal comes from bigger objects, like protein aggregates and not from exosomes. Until this is not formerly done, the text should acknowledge that at least 7% of the total signal does not correspond to EV bound CD63.

2. Isolation of EVs. The use of a distinct method of isolation for plasma EVs is not justified. Exoquick is known to co-isolate lipoprotein contaminants (Van Deun et al. JEV 2014).

In addition, this sentence: 'The transmission electron microscopy (TEM) displayed exosomes as vesicles with a lipid bilayer structure and sizes in the 50-150 nm range (Fig. 3a & d).' is wrong. Electron micrographs in Fig. 3a does not show a lipid bilayer.

Response to reviewer #1:

Q: 1. Specificity of the CD63-nanoLuc for the tracking of exosomes. The central point is the demonstration of the specificity of the signal. The separation of mouse plasma by molecular weight does not provide proofs that part of the signal comes from bigger objects, like protein aggregates and not from exosomes. Until this is not formerly done, the text should acknowledge that at least 7% of the total signal does not correspond to EV bound CD63.

R: We understand the concern and have included that "...7% was due to exosome-free NanoLuc..." in the manuscript as the reviewer suggested.

Q: 2. Isolation of EVs. The use of a distinct method of isolation for plasma EVs is not justified. Exoquick is known to co-isolate lipoprotein contaminants (Van Deun et al. JEV 2014).

R: Exoquick isolation, like any methodology, has its limitation. We understand the reviewer's concern. However, there is no single gold-standard separation method for exosome isolation including centrifugation. The key point is to carefully choose the isolation method that serves the best for the question of the study. As the Guidelines of International Society for Extracellular Vesicles indicates: "There is no single optimal separation method, so choose based on the downstream applications and scientific question." (*J Extracell Vesicles* **7**, 1535750 (2018)).

Since the purpose of isolated exosomes from mouse plasma is to assess plasma exosome morphology, expression of exosome markers, luciferase activities and nanoparticle tracking analysis, lipoprotein contaminants do not lead to the changes of these experimental assessments.

Q: 3. In addition, this sentence: 'The transmission electron microscopy (TEM) displayed exosomes as vesicles with a lipid bilayer structure and sizes in the 50-150 nm range (Fig. 3a & d).' is wrong. Electron micrographs in Fig. 3a does not show a lipid bilayer.

R: We appreciate the reviewer's point and have revised it accordingly.